# MTCH2 modulates CPT1 activity to regulate lipid metabolism of adipocytes

Chunyan Wu[1], Tongtong Wang[1], Adhideb Ghosh[1], Fen Long[1], Anand Kumar Sharma [1], Tina Dahlby[1], Falko Noé [1], Ilenia Severi[2], Georgia Colleluori[2], Saverio Cinti[2], Antonio Giordano[2], Lianggong Ding[1], Radhika Khandelwal[1], Sarantos Kostidis [3], Martin Giera [3], Lucia Balazova[4], Vincent Gardeux [5,6], Laith Abu-Nawwas[5,6], Bart Deplancke [5,6], Sabita Chourasia[7], Sandra Kleiner[8], Bradford S. Hamilton[8], Juan Manuel Alcántara Alcántara[9,10,11], Jonatan R. Ruiz [9,10,11], Matthias Blüher [12,13], Anton Pekcec [8], Miroslav Balaz[4,14], Atan Gross [7], Heike Neubauer [8] & Christian Wolfrum [1,15] ✉

Metabolic disorders, including obesity and metabolic-associated steatohepatitis, arise from a chronic energy surplus. Thus, enhancing energy dissipation through increased respiration holds significant therapeutic potential for metabolic disorders. Through a comprehensive analysis of human and murine adipose tissues, along with a functional screen, we identify mitochondrial carrier homolog 2, a mitochondrial outer membrane protein, as a pivotal regulator of mitochondrial metabolism. Intriguingly, its expression in adipose tissue is a strong determinant of obesity in humans. Adipocyte-specific ablation of mitochondrial carrier homolog 2 improves mitochondrial function and whole-body energy expenditure, independent of uncoupling protein 1. Furthermore, mitochondrial carrier homolog 2 regulates mitochondrial influx of free fatty acids by modulating the sensitivity of carnitine palmitoyltransferase 1 to malonyl-CoA through direct physical interaction, leading to enhanced energy expenditure in adipocytes/adipose tissue. Here we show mitochondrial carrier homolog 2 functions as a negative regulator of energy metabolism in adipocytes and represents a potential target for treating obesity and related metabolic disorders.

The global obesity epidemic is a driver of health burden, primarily due to its strong association with metabolic co-morbidities, such as type 2 diabetes mellitus, and cardiovascular diseases[1]. As a major endocrine organ, adipose tissue is organized in different anatomical depots, discretely located all over the body. Brown adipose tissue (BAT), along with functionally related beige adipocytes, dissipates chemical energy in the form of heat[2,3]. This metabolic heat generation protects against cold via non-shivering thermogenesis and can potentially counteract energy overload[2,3]. Conversely, white adipose tissue (WAT) serves as an endocrine organ to store energy. WAT stores the bulk of triacylglycerols and, upon stimulation, acutely mobilizes fatty acids (FAs) for energy production through lipolysis[4]. Both BAT and WAT are critical for maintaining whole-body energy balance and regulating glucose and lipid metabolism, while its dysfunction is etiologically implicated in obesity and diabetes[5-7]. Therefore, promoting energy expenditure is a promising anti-obesity approach. The uncoupling protein 1 (UCP1)-mediated thermogenesis in BAT and beige adipocytes is an extensively studied energy-dissipating process[3]. However, adult humans exhibit a moderate expression of UCP1[8-10]. Nonetheless, emerging evidence suggests that adipose tissues can also contribute to

energy expenditure through UCP1-independent mechanisms[11–13]. These findings mostly emphasize the positive modulators of energy expenditure. Intuitively, negative regulators of energy expenditure could be an attractive target, yet remain poorly studied.

Mitochondria are pivotal in cellular energy production and dissipation. They utilize glucose or FAs as primary fuel sources involving complex pathways that serve as substrates for mitochondrial respiration, which is crucial for maintaining cellular energy balance and metabolic stability[14,15]. Several key enzymes govern the rate-limiting steps in fuel breakdown for mitochondrial utilization. In the context of lipid metabolism, lipases hydrolyze triglycerides to generate FAs as primary usable fuel, while carnitine palmitoyltransferase 1 (CPT1) acts as a gatekeeper regulating FAs entry into mitochondria[14]. The availability of substrates and especially FAs to mitochondria serves as a limiting factor for mitochondrial function in energy production[15,16]. Therefore, altered availability or the activity of these enzymes can drastically impact metabolic flux and energy expenditure.

Mitochondrial carrier homolog 2 (MTCH2) localizes on the outer membrane of mitochondria. It belongs to the solute carrier 25 family,

but its substrates or transporter function remain unknown[17]. Previous studies have implicated MTCH2 in various physiological processes, including apoptosis induction[18–20], mitochondrial metabolism and dynamics[21–24], and its role as a protein insertase[25] and scramblase[26]. Moreover, MTCH2 is an important regulator of adipocyte differentiation and lipid homeostasis[23,27–32]. Genetic alterations in *MTCH2* have been linked to diverse disease phenotypes, including obesity, Alzheimer's disease, and cancer[33]. Notably, genome-wide association studies have identified *MTCH2* variants associated with increased obesity and diabetes risk[34]. However, how MTCH2 regulates these diverse molecular functions and how it relates to its pathological association remains unclear. While MTCH2 emerges as a promising candidate for modulating adipocyte function and whole-body energy metabolism, its adipose-specific function remains unknown. In this study, we show that MTCH2 regulates mitochondrial function and lipid metabolism by controlling mitochondrial influx of FAs through modulation of CPT1 sensitivity to malonyl-CoA.

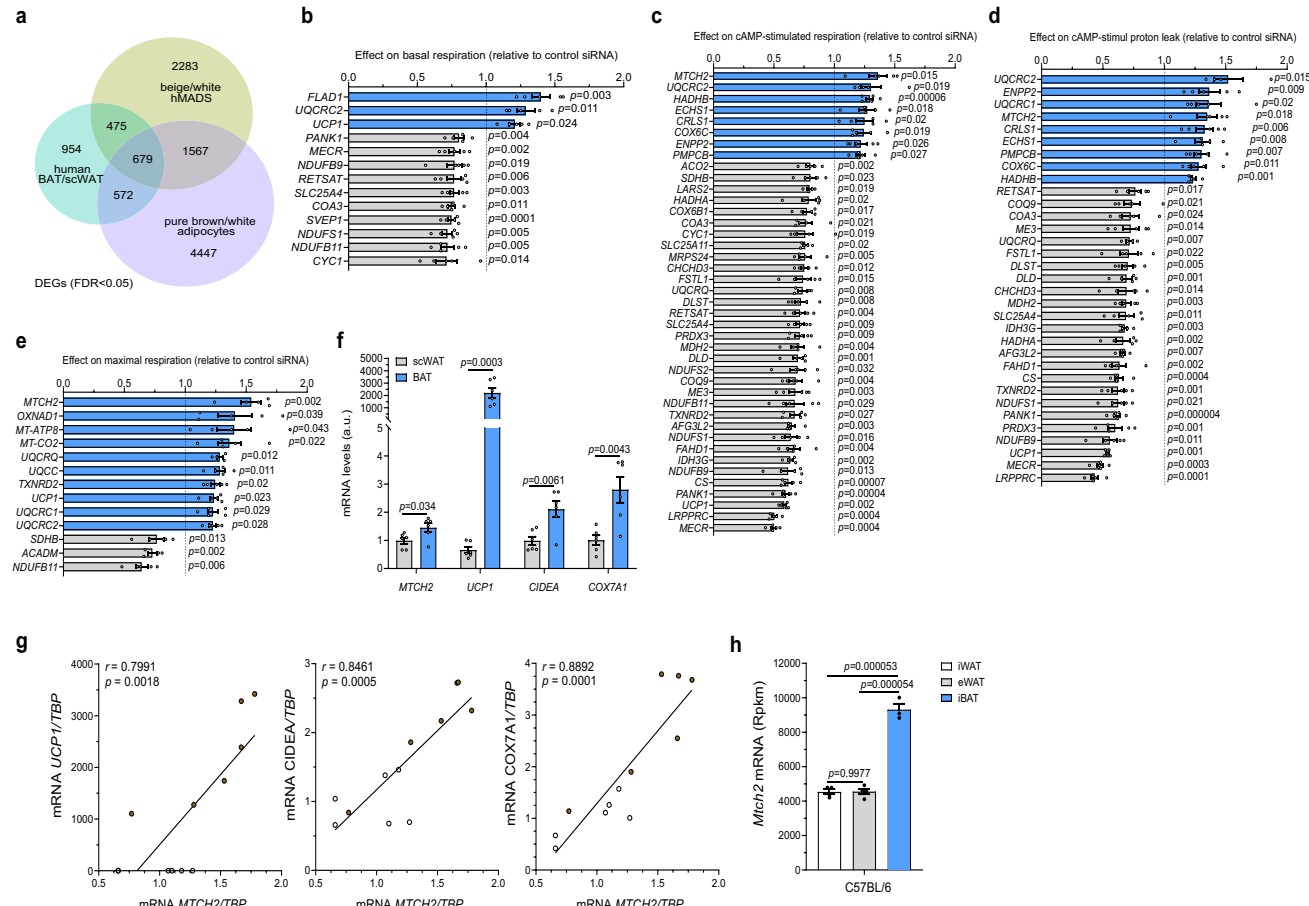

Fig. 1 | Identification of targets with potential to promote adipocyte function. **a** Identification of 679 overlapping differentially expressed genes (DEGs; FDR adjusted p < 0.05) by cross-analyzing transcriptomes of paired supraclavicular brown adipose tissue (BAT) and adjacent subcutaneous white adipose tissue (scWAT) biopsies (n = 6, in green), human multipotent adipose-derived stem (hMADS) cells differentiated into white and beige adipocytes (n = 3, in yellow) and pure murine brown and white adipocytes (n = 5, in purple). **b–e** Mitochondrial respiration in hMADS differentiated beige adipocytes following knockdown of 110 candidate genes. Positive effects on mitochondrial respiration are depicted in blue, negative effects in gray. Genes significantly affecting **b** basal (n = 5 for each group), **c** dibutyryl cyclic adenosine monophosphate (cAMP)-stimulated (n = 4 for *UQCRQ* group and n = 5 for other groups), **d** cAMP-stimulated uncoupled (n = 4 for *UQCRQ*,

*UQCRC2* groups and n = 5 for other groups), and **e** maximal mitochondrial respiration (n = 4 for *UQCRQ, ACADM* groups and n = 5 for other groups) by more than 20% are shown. **f** Mitochondrial carrier homolog 2 (*MTCH2*) expression levels in human scWAT (in gray) and supraclavicular BAT (in blue) biopsies. n = 6 for each group. **g** The correlation of *MTCH2* expression with *UCP1, CIDEA*, and *COX7A1* in human scWAT and supraclavicular BAT biopsies. n = 12 for each group. **h** *Mtch2* expression levels in mouse inguinal WAT (iWAT, in white), epididymal WAT (eWAT, in gray), and BAT (in blue). n = 4 for iWAT and eWAT, and n = 3 for BAT. Data are presented as mean ± SEM and analyzed using two-tailed Student's *t*-test (**b**–**f**) or a two-tailed Pearson test (**g**), or two-way ANOVA with Tukey's post hoc multiple comparison test (**h**). Source data are provided as a Source data file.

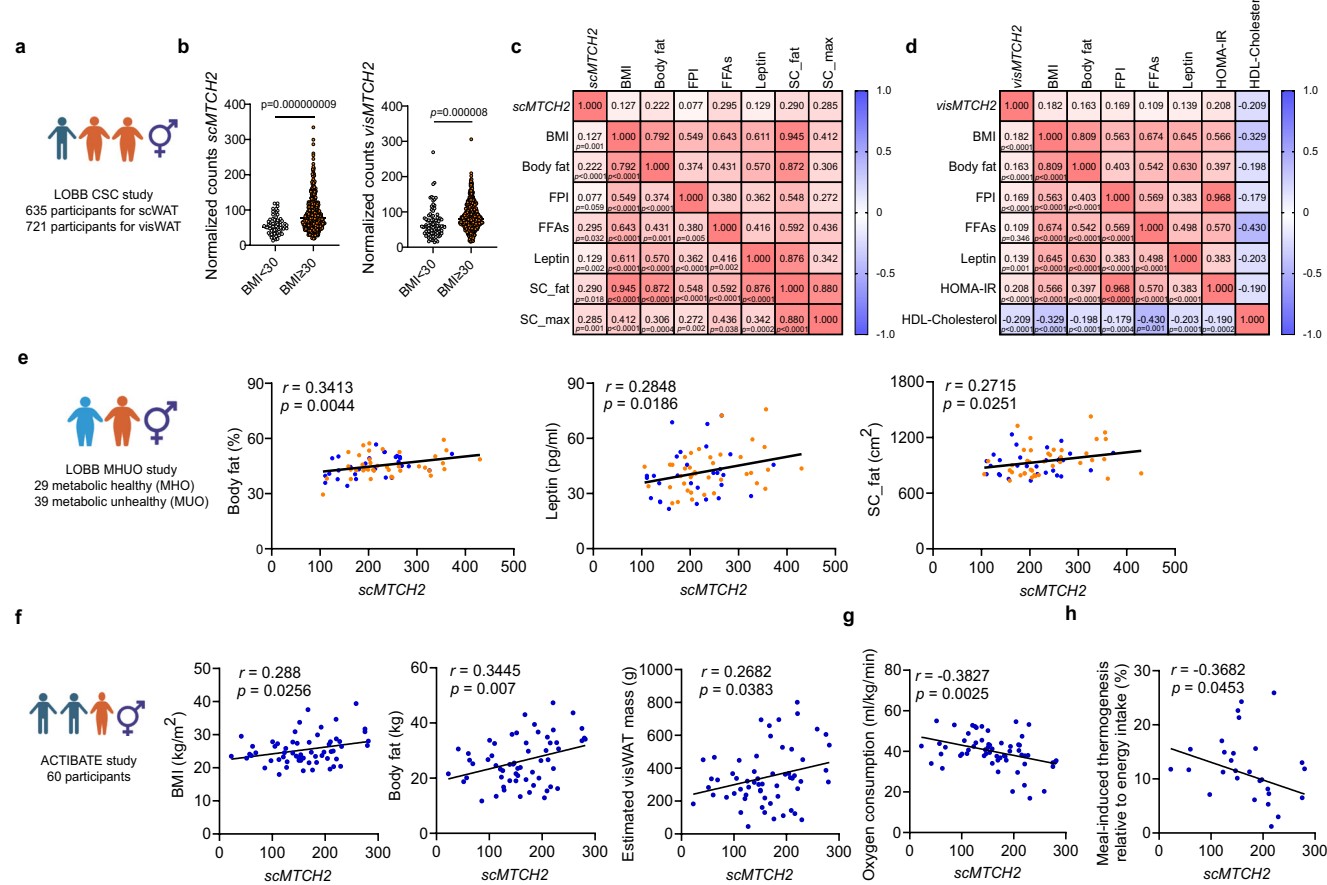

**Fig. 2 | Mitochondrial carrier homolog 2 (MTCH2) expression in humans is linked to a metabolic profile associated with obesity. a** Leipzig Obesity BioBank (LOBB) cross-sectional cohort (CSC) study, integrating clinical parameters with bulk RNA sequencing of subcutaneous white adipose tissue (scWAT) and visceral WAT (visWAT) (created in BioRender, https://BioRender.com/s1eg8w0). **b** MTCH2 expression between lean (body mass index (BMI) < 30, white circles) individuals and individuals living with obesity (BMI ≥ 30, orange circles) in scWAT (scMTCH2; n = 70 and 565 individuals, respectively) and in visWAT (visMTCH2; n = 96 and 625 individuals, respectively). **c** Correlation of scMTCH2 expression with various clinical parameters. BMI (n = 635); body fat (n = 613); FPI (fasting plasma insulin; n = 608); FFAs (free fatty acids; n = 53); leptin (n = 545); SC_fat (subcutaneous fat area; n = 66); SC_max (subcutaneous max adipocyte size; n = 130). **d** Correlation of visMTCH2 expression with various clinical parameters. BMI (n = 721); body fat (n = 691); FPI

(n = 683); FFAs (n = 77); leptin (n = 625); HOMA-IR (homeostatic model assessment for insulin resistance; n = 615); HDL-cholesterol (high-density lipoprotein-cholesterol; n = 404). **e** LOBB metabolically healthy and unhealthy (MHUO) study (created in BioRender, https://BioRender.com/esvem6q). Correlation of scMTCH2 expression with body fat, leptin, and SC_fat in metabolically healthy (in blue; n = 29 individuals) versus unhealthy (in orange; n = 39 individuals) obesity cohorts.
**f–h** ACTIBATE young human study (created in BioRender, https://BioRender.com/z9lbecl). Correlation of scMTCH2 expression with **f** BMI, body fat, and estimated visWAT mass (n = 60 individuals), **g** relative oxygen consumption levels (n = 60 individuals), and **h** meal-induced thermogenesis relative to energy intake (n = 30 individuals). Data are presented as mean ± SEM and analyzed using two-tailed Student's t-test (**b**), or a two-tailed Pearson test (**c–h**). The exact p values for (**c, d**) are shown in the Source data file. Source data are provided as a Source data file.

## Results

### Identifying potential targets to promote adipocyte function

Mitochondrial respiration/metabolism regulators are attractive targets to increase energy expenditure for metabolic benefits. In order to identify potential target genes with the potential to promote adipocyte function, we conducted a cross-analysis of transcriptomes from various sources, including (i) paired supraclavicular BAT and adjacent subcutaneous WAT (scWAT) biopsies of six healthy lean volunteers, (ii) human multipotent adipose-derived stem (hMADS) cells differentiated into white and beige adipocytes, as well as (iii) pure murine interscapular brown (iBAT) and inguinal white (iWAT) populations isolated via fluorescence-activated cell sorting. Comparing the transcriptomes of human BAT and scWAT biopsies, we identified 2680 differentially expressed genes (DEGs) (Fig. 1a and Supplementary Table 1). Analyzing transcriptomes of white and beige hMADS adipocytes, we identified 5004 DEGs, with 2755 upregulated and 2249 downregulated genes in beige adipocytes (Fig. 1a and Supplementary Table 2). Additionally,

comparison of pure murine iBAT and iWAT populations revealed significant enrichment in the expression of 3305 genes in brown and 3960 genes in white adipocytes, respectively, resulting in a total of 7265 DEGs (Supplementary Table 3). Cross-analysis of these three DEG datasets (human BAT vs. scWAT, white vs. beige hMADS adipocytes, and murine iBAT vs. iWAT) revealed 679 overlapping DEGs (FDR < 0.05; Fig. 1a and Supplementary Table 4). Notably, our DEGs list includes multiple known BAT markers and regulators of metabolic activity, such as UCP1, GYK, CPT1, COX7A1, and CIDEA, confirming the accuracy of our transcriptomic analysis (Supplementary Table 4).

### MTCH2 modulates cellular respiration in human adipocytes

From the list of DEGs, we performed a knowledge-guided candidate selection (depot-specific expression, known molecular function, and physiological correlations), and selected 110 most promising genes for functional validation (Supplementary Table 5). We first conducted a short interfering RNAs (siRNAs) functional screen to identify genes

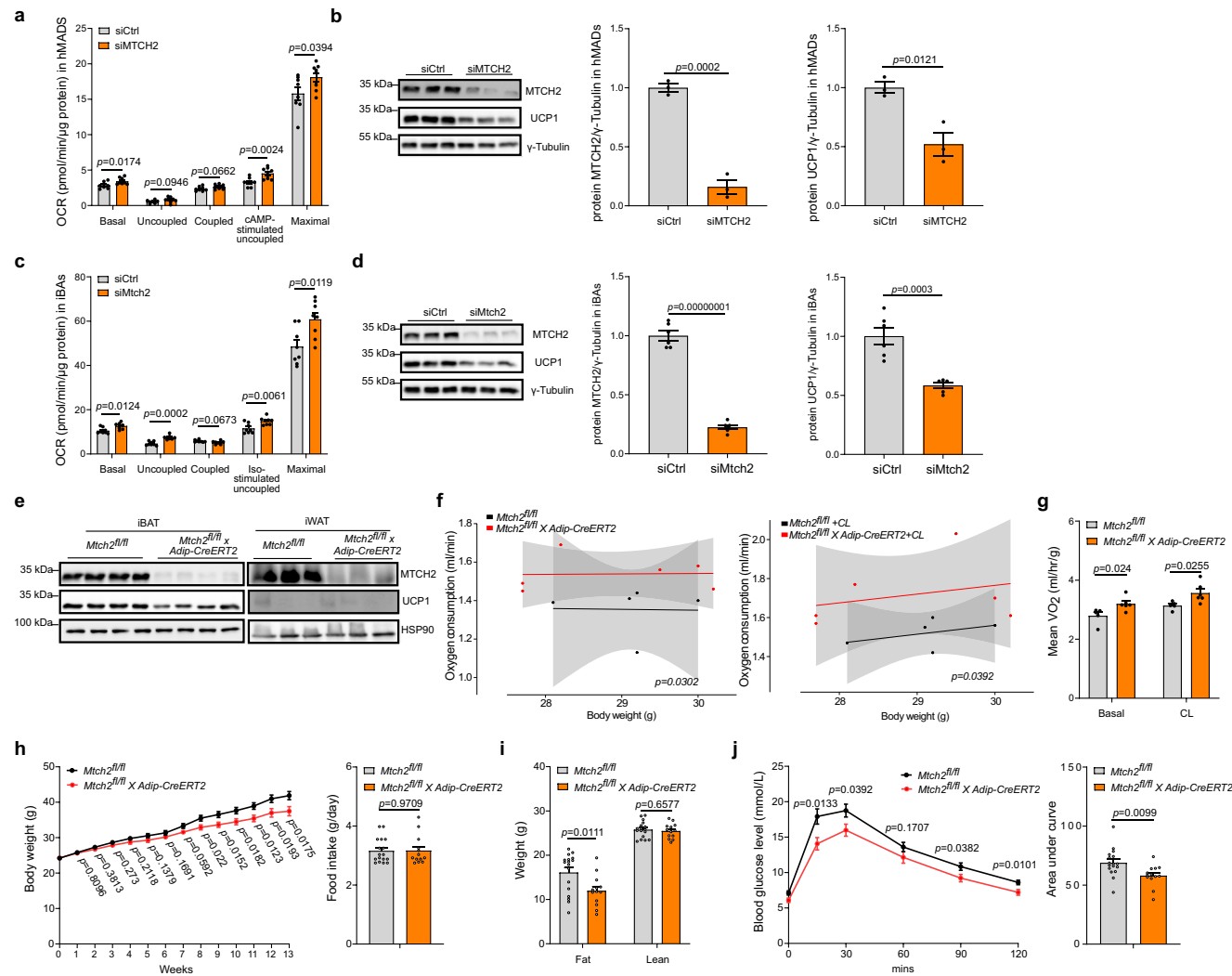

**Fig. 3 | Mitochondrial carrier homolog 2 (MTCH2) ablation enhances mitochondrial respiration in vitro and energy expenditure in vivo. a** The effects of *MTCH2* knockdown (KD) on oxygen consumption rate (OCR) in human multipotent adipose-derived stem (hMADS) adipocytes. n = 8 for control group (siCtrl) in gray and n = 9 for *MTCH2* KD group (siMTCH2) in orange. cAMP, dibutyryl cyclic adenosine monophosphate. **b** The effects of *MTCH2* ablation on uncoupling protein 1 (UCP1) protein levels (n = 3 for each group, control group in gray and *MTCH2* KD group in orange) in hMADS adipocytes. **c** The effects of *Mtch2* ablation on mitochondrial respiration (n = 8 for each group, control group in gray and *Mtch2* KD group in orange) in mature murine immortalized brown adipocytes (iBAs). Iso isoproterenol. **d** The effects of *Mtch2* ablation on UCP1 protein levels (n = 6 for each group, control group in gray and *Mtch2* KD group in orange) in iBAs. **e** Representative blots of MTCH2 and UCP1 protein levels in adipose tissue-specific *Mtch2* knockout (KO) mice

(*Mtch2*[fl/fl]*Adip-CreERT2*; n = 4 for iBAT and n = 3 for iWAT). **f, g** Energy expenditure in mice. **f** Analysis of covariance (ANCOVA) of oxygen consumption/body weight (control group mice n = 5 in black; *Mtch2* KO mice n = 6 in orange). **g** Average oxygen consumption under basal and CL-316,243 (CL, a thermogenic beta 3-agonist) injected conditions (control group mice n = 5 in gray; *Mtch2* KO mice n = 6 in orange). **h** Mice body weight and food intake on a high-fat diet (HFD; control group mice n = 17 in gray; *Mtch2* KO mice n = 13 in orange). **i** Mice body composition analysis using the EchoMRI after 13 weeks on an HFD (control group mice n = 17 in gray; *Mtch2* KO mice n = 13 in orange). **j** Intraperitoneal glucose tolerance test and area under the curve after 12 weeks on an HFD (control group mice n = 17 in gray; *Mtch2* KO mice n = 13 in orange). Data are presented as mean ± SEM and analyzed using two-tailed Student's *t*-test (**a**–**d**, **g**–**j**) or two-tailed ANCOVA (**f**). Source data are provided as a Source data file.

involved in the regulation of hMADS adipocytes respiration. Among these, the knockdown of 13 genes significantly affected basal mitochondrial respiration by at least 20% (Fig. 1b). Subsequent injection of dibutyryl cyclic adenosine monophosphate (cAMP), a stable cAMP analog, to stimulate mitochondrial function in adipocytes, led to the identification of 41 genes whose knockdown significantly altered cAMP-stimulated mitochondrial respiration (≥20% effect; Fig. 1c) in hMADS adipocytes. Notably, 8 genes caused a significant increase in cAMP-stimulated mitochondrial respiration, including *CRLS1* (Fig. 1c), which has previously been shown to modulate adipocyte metabolic activity[35].

Next, oligomycin injection was performed to block ATP synthase, enabling us to assess the contribution of proton leak to mitochondrial

respiration. This analysis revealed that the knockdown of 9 genes significantly promoted cAMP-stimulated proton leak, while the knockdown of other 24 genes inhibited it (≥20% effect; Fig. 1d). As expected, the knockdown of *UCP1*, which encodes the key effector protein of mitochondrial uncoupling, exhibited one of the strongest inhibitory effects on cAMP-stimulated proton leak (Fig. 1d). Finally, we used FCCP, a chemical uncoupler, to calculate the maximal mitochondrial respiration capacity. The knockdown of 10 genes resulted in a significant increase in maximal mitochondrial respiration, while 3 genes showed a reduced respiration capacity (≥20% effect; Fig. 1e). Combining the results from these investigations, we identified several genes implicated in the regulation of mitochondrial function in mature human hMADS adipocytes. Notably, among the genes whose

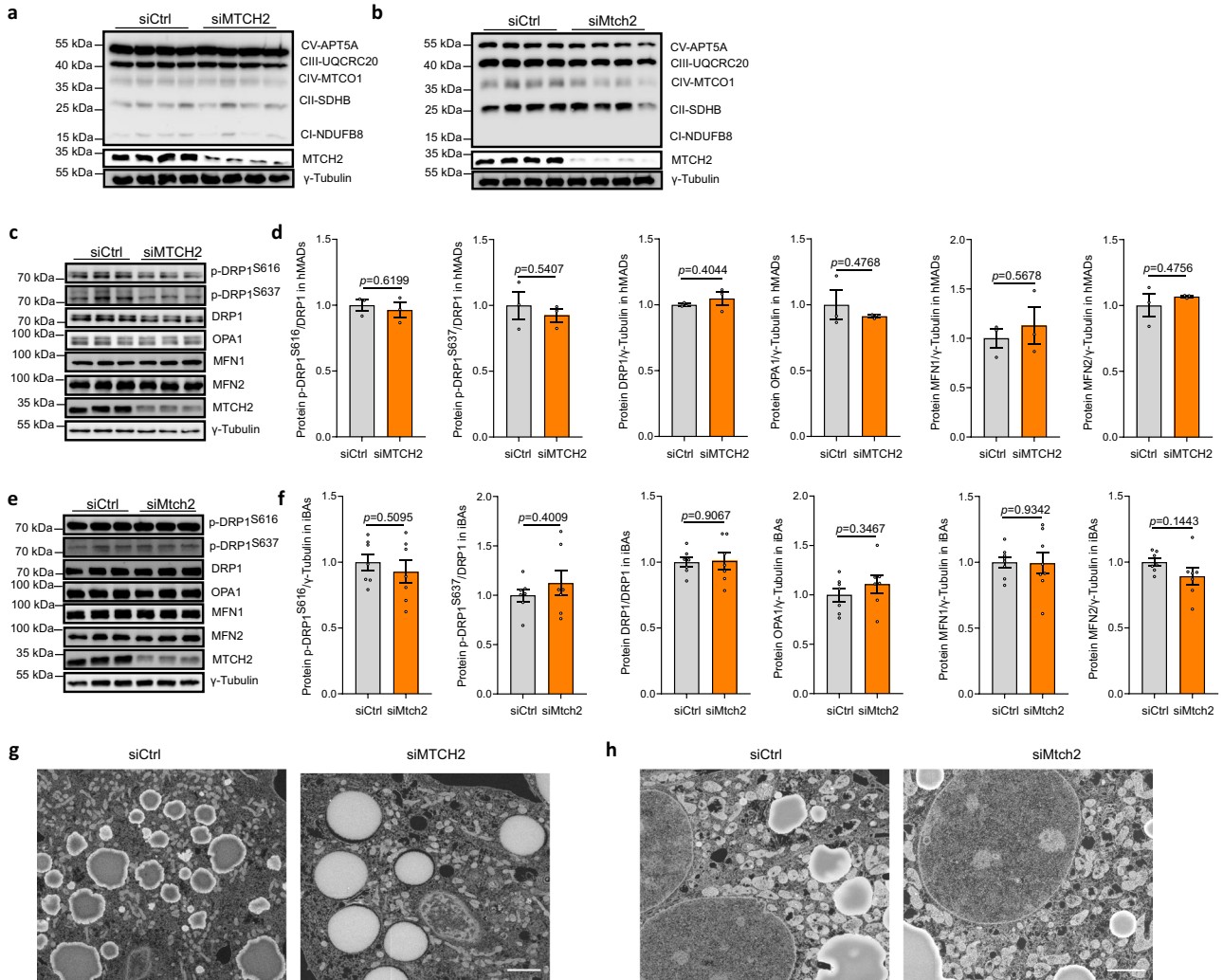

**Fig. 4 | Mitochondrial carrier homolog 2 (MTCH2) regulates mitochondrial metabolism independently of mitochondrial dynamics in human adipocytes.** The effects of MTCH2 knockdown (KD) on cellular levels of individual oxidative phosphorylation (OXPHOS) proteins in **a** human multipotent adipose-derived stem (hMADS) adipocytes and **b** mature murine immortalized brown adipocytes (iBAs). n = 4 for each group, control group (siCtrl) in gray and MTCH2 KD group (siMTCH2) in orange. Representative blots (**c**) and quantification (**d**) of core indicated mitochondrial dynamics proteins in hMADS adipocytes with MTCH2 KD (n = 3 for each group, control group in gray and MTCH2 KD group in orange). Representative blots

(**e**) and quantification (**f**) of core indicated mitochondrial dynamics proteins in iBAs with Mtch2 KD. DRP1 dynamin-related protein 1, OPA1 optic atrophy 1, MFN1 mitofusin-1, MFN2 mitofusin-2. n = 8 for MFN1, and n = 7 for all other groups (control group in gray and Mtch2 KD group in orange). **g, h** The effects of Mtch2 ablation on mitochondrial morphology in hMADS adipocytes and iBAs. Representative electron microscopy (EM) images are presented **g** in hMADS adipocytes and **h** iBAs. Scale bar, 5 μm. Experiments were repeated two times. Data are presented as mean ± SEM and analyzed using a two-tailed Student's t-test. Source data are provided as a Source data file.

knockdown significantly increased individual parameters of mitochondrial respiration, MTCH2 demonstrated the strongest effects.

## MTCH2 expression is correlated with mitochondrial function
The MTCH2 transcript exhibited significant abundance in human supraclavicular BAT and scWAT biopsies, with higher levels observed in BAT (Fig. 1f and Supplementary Fig. 1a). This expression pattern of MTCH2 correlated with the levels of genes involved in mitochondrial function in human adipose tissues (Fig. 1g), and is predominantly observed in the human adipocyte population, as revealed by single-nucleus RNA sequencing data (Supplementary Fig. 1b). Consistent with this finding, MTCH2 was expressed in both hMADS white and beige adipocytes, with higher levels observed in beige adipocytes (Supplementary Fig. 1c, d). To further assess the expression pattern of Mtch2, we analyzed its mRNA levels in different adipose tissue depots in mice. We found that Mtch2 is expressed in both WAT and iBAT, with relatively higher levels observed in iBAT compared to iWAT and

epididymal WAT (eWAT) in C57BL/6 mice (Fig. 1h). Interestingly, even in Ucp1-negative white adipocytes, Mtch2 levels were substantial in both female and male C57BL/6 mice (Supplementary Fig. 1e). Moreover, Mtch2 expression is upregulated by high-fat diet (HFD) and downregulated under thermoneutral conditions (Supplementary Fig. 1f). Analysis of published datasets[36,37] demonstrated that its expression is induced by cold exposure and decreased under thermoneutrality (Supplementary Fig. 1g). These data collectively suggest that MTCH2 is expressed across both brown and white adipocytes, with a higher expression in brown adipocytes correlated with the degree of mitochondrial catabolism. Notably, this expression pattern is consistent between humans and mice, reinforcing the translational relevance of these findings.

## MTCH2 is linked to obesity-related metabolism in humans
Next, we analyzed the expression of MTCH2 in well-characterized cohorts with detailed phenotypic data to study its metabolic relevance

in humans. Initially, we examined *MTCH2* expression in a large LOBB cohort (45.4 ± 13.8 years old; body mass index (BMI): 45.2 ± 10.9 kg/m²; Fig. 2a). The *MTCH2* expression was significantly higher in both scWAT and visceral WAT (visWAT) from individuals living with obesity (BMI ≥ 30) compared to lean individuals (BMI < 30) (Fig. 2b), suggesting the relevance of MTCH2 to WAT in the context of obesity, which is consistent with previous studies[34,38]. To further explore the relationship between MTCH2 and obesity, we analyzed the correlations between *MTCH2* expression and various clinical parameters. In human scWAT, we observed a positive correlation of *MTCH2* expression (*scMTCH2*) with several parameters, including BMI, body fat, FFAs, leptin, subcutaneous fat area (SC-fat), and SC_max (Fig. 2c and Supplementary Fig. 2a, b). In human visWAT, we observed a positive correlation of *MTCH2* expression (*visMTCH2*) with BMI, body fat, fasting plasma insulin (FPI), leptin, homeostatic model assessment for insulin resistance (HOMA-IR), and HDL-cholesterol (Fig. 2d and Supplementary Fig. 2c, d). We then examined another subcohort of LOBB study of individuals living with obesity, further distinguishing those with metabolically healthy obesity (MHO) and metabolically unhealthy obesity (MUO)[39] (43 ± 10 years old; BMI: 46.3 ± 7.1 kg/m²; Fig. 2e, left panel). In this group, *MTCH2* expression was also positively correlated with body fat, leptin, and SC-fat (Fig. 2e), in line with findings from our larger LOBB cohort. Additionally, we assessed *MTCH2* expression in young human scWAT from the ACTIBATE cohort (22.1 ± 2.1 years old; BMI: 25.5 ± 4.4 kg/m²; Fig. 2f). We also observed that *scMTCH2* expression was positively correlated with BMI, body fat, visWAT mass and total fat mass percentage (Fig. 2f and Supplementary Fig. 2e). Most importantly, *scMTCH2* expression was negatively correlated with oxygen consumption rate (Fig. 2g) and meal-induced thermogenesis (Fig. 2h). These findings support the physiological validity of our screen data and establishes MTCH2 as a potential modulator of obesity and energy metabolism.

## MTCH2 regulates energy expenditure independent of UCP1

Next, we sought to dissect the mechanistic underpinnings of the MTCH2-mediated regulation of metabolism. In line with our initial screen findings, we validated that *MTCH2* knockdown (Supplementary Fig. 3a) significantly increased basal, cAMP-stimulated uncoupled, and maximal mitochondrial respiration in hMADS adipocytes (Fig. 3a). Notably, this stimulatory effect was in the backdrop of a significant reduction in UCP1 protein level (Fig. 3b and Supplementary Fig. 3a), without affecting major adipogenic and thermogenic marker genes, including *FABP4*, *ADIPOQ*, *PPARG*, *PPARGC1A* and *CIDEA* (Supplementary Fig. 3b). We also reproduced the respiratory phenotype in mature murine immortalized brown adipocytes (iBAs) that recapitulated the knockdown effect (Supplementary Fig. 3c). *Mtch2* knockdown markedly elevated basal, uncoupled, isoproterenol-stimulated uncoupled, and maximal mitochondrial respiration (Fig. 3c), which was also associated with a pronounced reduction in UCP1 protein levels (Fig. 3d). To further consolidate the UCP1 independence of *Mtch2* knockdown-induced increase in respiration, we performed dual knockdown of *Mtch2* and *Ucp1*. The effects of *Mtch2* on cellular respiration persisted, even in the *Ucp1*-ablated iBAs and primary brown adipocytes (Supplementary Fig. 3d, e). These findings suggest that MTCH2 efficiently modulates adipocyte respiration, independent of UCP1.

To investigate the physiological significance of MTCH2 in mice, we subsequently generated adipocyte-specific *Mtch2* knockout mice (*Mtch2*^fl/fl^*Adip-CreERT2*; Fig. 3e). The physical appearance, birth rate, and growth of *Mtch2*^fl/fl^*Adip-CreERT2* mice appeared to be normal. Following this, we subjected these mice to comprehensive metabolic characterization. When fed a regular chow diet, *Mtch2*^fl/fl^*Adip-CreERT2* mice exhibited a similar pattern of weight gain and food intake compared to control littermates (Supplementary Fig. 3f). However, *Mtch2*^fl/fl^*Adip-CreERT2* mice demonstrated an increase in energy expenditure

accompanied with a reduction in respiratory exchange ratio, without affecting food intake and locomotor activity (Fig. 3f, g and Supplementary Fig. 3g, h). This enhanced energy expenditure prompted us to investigate their response to an HFD. When subjected to HFD, *Mtch2*^fl/fl^*Adip-CreERT2* mice exhibited reduced susceptibility to weight gain compared to control littermates, while maintaining similar food intake (Fig. 3h). Body composition analysis using EchoMRI revealed that *Mtch2*^fl/fl^*Adip-CreERT2* mice had lower fat mass and similar lean mass compared to controls (Fig. 3i). Additionally, *Mtch2*^fl/fl^*Adip-CreERT2* mice demonstrated improved glucose tolerance on the HFD (Fig. 3j). Plasma triglyceride levels remained unchanged, whereas plasma cholesterol levels were significantly decreased in *Mtch2*^fl/fl^*Adip-CreERT2* mice (Supplementary Fig. 3i). Collectively, these results demonstrate that the absence of adipocyte MTCH2 protects mice from HFD-induced obesity by enhancing energy expenditure.

## MTCH2 functions independent of cell apoptosis and OXPHOS

A recent study suggested that dying neighboring cells secrete inosine that could increase energy expenditure[40]. Incidentally, MTCH2 has been implicated in regulating cell apoptosis and mitochondrial function, including OXPHOS complexes in various cell types and tissues[18–20,31,41]. Thus, we sought to check the possibility of a cell death-mediated increase in energy expenditure in the MTCH2-deleted adipocytes. We quantified the apoptotic effect of siRNA-mediated knockdown of *MTCH2* in both hMADS adipocytes and iBAs. However, the cell viability of both hMADS adipocytes and iBAs remained unaltered (Supplementary Fig. 4a, b), suggesting that the effects of MTCH2 on adipocytes are independent of apoptosis. Similarly, we could not detect any upregulation of individual OXPHOS proteins following *MTCH2* silencing in either hMADS adipocytes (Fig. 4a) or iBAs (Fig. 4b). In alignment with our in vitro data, the levels of individual OXPHOS proteins remained unchanged in iBAT of *Mtch2*^fl/fl^*Adip-CreERT2* mice (Supplementary Fig. 4c). These data suggest that the respiration/energy expenditure phenotype upon loss of MTCH2 is not due to cell apoptosis or OXPHOS complexes levels.

## MTCH2 functions independent of mitochondrial dynamics

MTCH2 has been reported to modulate the dynamics of mitochondrial biogenesis. It was noted that MTCH2 regulates mitochondrial elongation in stem cells[22], and acts as a selective mediator of starvation-induced mitochondrial hyperfusion[23]. We, therefore, wondered if this could be the mechanism by which loss of MTCH2 affects systemic energy homeostasis. However, under both basal and stimulated conditions, the loss of *MTCH2* did not cause any significant changes in the expression levels of core mitochondrial dynamics proteins, including mitofusins (MFN1 and MFN2), optic atrophy 1 (OPA1), dynamin-related protein 1 (DRP1) and its phosphorylation levels, in both hMADS adipocytes and iBAs under basal (Fig. 4c–f) and stimulated conditions (Supplementary Fig. 4d, e).

We also performed electron microscopy imaging to characterize the effect of MTCH2 on mitochondrial morphology in both hMADS adipocytes and iBAs. No obvious changes were observed in mitochondrial morphology in hMADS adipocytes (Fig. 4g) and iBAs (Fig. 4h). Taken together, these results suggest that MTCH2 regulates mitochondrial metabolism independently of the mitochondrial fission/fusion process in both human and mouse adipocytes. This observation implies that the metabolic effects of MTCH2 in cultured human and mouse adipocytes are mediated by mechanisms that remain to be uncovered.

## MTCH2 deficiency increases FA oxidation (FAO) in adipocytes

The observation of enhanced energy expenditure without a corresponding increase in mitochondrial OXPHOS protein levels with *MTCH2* ablation is indeed a notable finding. To uncover the underlying mechanisms driving this discrepancy, we focused on investigating

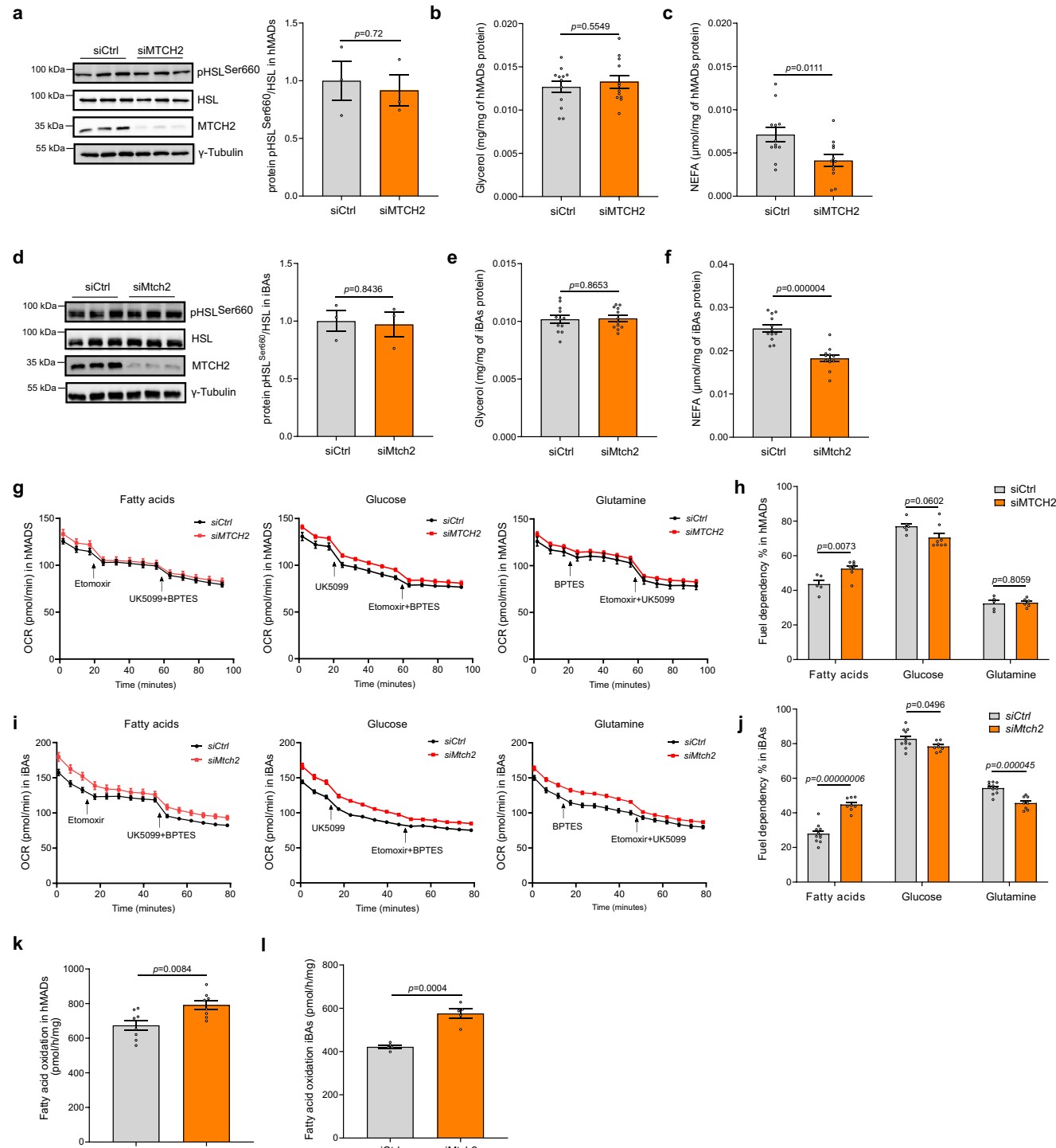

**Fig. 5 | Mitochondrial carrier homolog 2 (MTCH2) deficiency increases fatty acid oxidation (FAO) in adipocytes. a** Representative blots and quantification of hormone-sensitive lipase (HSL) and its phosphorylation levels in human multipotent adipose-derived stem (hMADS) adipocytes with *MTCH2* knockdown (KD). n = 3 for each group, control group (siCtrl) in gray and *MTCH2* KD group (siMTCH2) in orange. Glycerol (**b**) and non-esterified fatty acids (NEFA) (**c**) levels in the medium of hMADS adipocytes with *MTCH2* KD (n = 12 for each group, control group in gray and *MTCH2* KD group in orange). **d** Representative blots and quantification of HSL and its phosphorylation levels in mature murine immortalized brown adipocytes (iBAs) with *Mtch2* KD (n = 3 for each group, control group in gray and *Mtch2* KD group in orange). Glycerol (**e**) and NEFA (**f**) levels in the medium of iBAs with *mtch2* KD (n = 12 for each group, control group in gray and *Mtch2* KD group in

orange). Fuel dependency assay by oxygen consumption rate (OCR) (**g**) and its quantification (**h**) in hMADS adipocytes with *MTCH2* KD. FA dependency assay, n = 5 for control group in gray and n = 8 for *MTCH2* KD group in orange. Glucose dependency assay, n = 6 for control group in gray and n = 8 for *MTCH2* KD group in orange. Glutamine dependency assay, n = 5 for control group in gray and n = 6 for *MTCH2* KD group in orange. Fuel dependency assay (**i**) and its quantification (**j**) in iBAs with *mtch2* KD (n = 11 for control group in gray and n = 8 for *Mtch2* KD group in orange). **k** The FAO rates in hMADS adipocytes with *MTCH2* KD (n = 8 for each group, control group in gray and *MTCH2* KD group in orange). **l** The FAO rate in iBAs with *mtch2* KD (n = 5 for each group, control group in gray and *Mtch2* KD group in orange). Data are presented as mean ± SEM and analyzed using a two-tailed Student's *t*-test. Source data are provided as a Source data file.

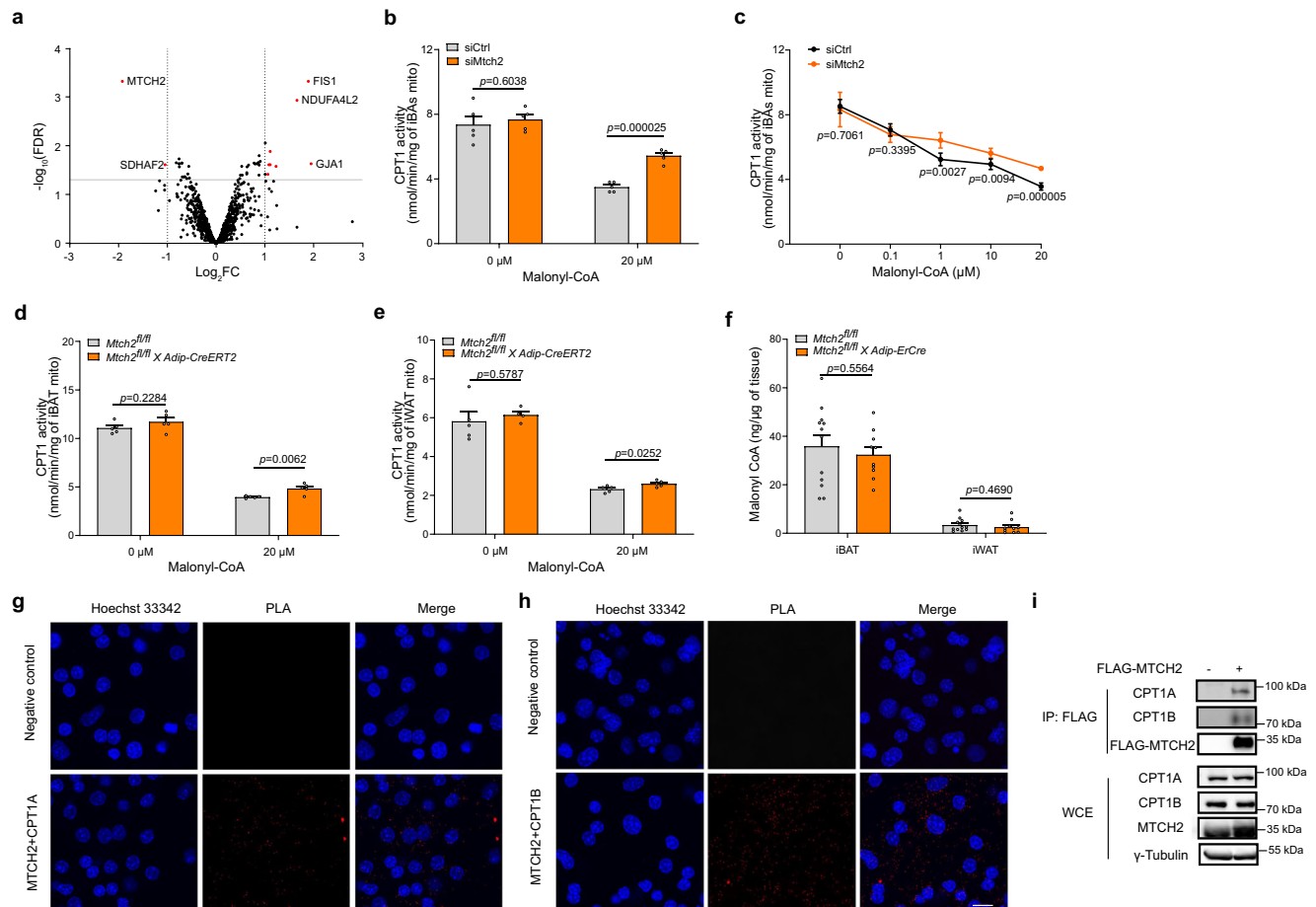

**Fig. 6 | Mitochondrial carrier homolog 2 (MTCH2) interacts with carnitine palmitoyltransferase 1 (CPT1) to modulate its activity and regulate fatty acid oxidation (FAO) in adipocytes. a** Proteomics of mitochondria isolated from control and *Mtch2* knockdown (KD) mature murine immortalized brown adipocytes (iBAs) (n = 4 for each group, differentially expressed proteins are highlighted in red). **b, c** CPT1 activities in the presence of different concentrations of malonyl-CoA in iBAs mitochondria. n = 5 for each group, control group (siCtrl) in gray and *Mtch2* KD (siMtch2) group in orange. CPT1 activities in the presence of different concentrations of malonyl-CoA in **d** interscapular brown adipose tissue (iBAT) and **e** inguinal white adipose tissue (iWAT) mitochondria isolated from *Mtch2^{fl/fl}Adip-*

*CreERT2* knockout (KO in orange) and control littermates in gray. n = 4 for iWAT from *Mtch2* KO mice at 0 μM malonyl-CoA, and n = 5 for other groups. **f** Malonyl-CoA levels in iBAT and iWAT from *Mtch2^{fl/fl}Adip-CreERT2* (n = 10 in orange) and control littermates (n = 12 in gray). **g, h** Representative images of proximity ligation assay (PLA). Direct interaction between MTCH2 and CPT1A (**g**), and CPT1B (**h**) in iBAs. Scale bar, 20 μm. Experiments were repeated three times. **i** Co-immunoprecipitation conducted in HEK293T whole cell extract (WCE) via FLAG antibody. Experiments were repeated three times. Data are presented as mean ± SEM and analyzed using a two-tailed Student's *t*-test. Source data are provided as a Source data file.

whether MTCH2 functions by regulating lipid metabolism in adipocytes. Thus, we examined the impact of MTCH2 on lipolysis, a critical process that regulates lipid utilization and energy balance in adipocytes. Silencing *MTCH2* did not alter hormone-sensitive lipase (HSL) phosphorylation levels in hMADS adipocytes (Fig. 5a), indicating no change in lipolysis. This finding was supported by the observation of unaltered glycerol release (Fig. 5b). However, we observed a significant decrease in FFAs levels in the media (Fig. 5c), suggesting enhanced utilization of FFAs in mature human adipocytes. Similar findings were observed in iBAs (Fig. 5d–f).

We thus investigated whether MTCH2 regulates lipid or glucose metabolization to enhance FFAs utilization, and therefore, we conducted a fuel dependency test, which utilizes specific combinations of enzyme inhibitors to assess the dependency of cells on three primary fuel sources (pyruvate, glutamine, or FAs) for mitochondria. In response to the loss of *MTCH2* in hMADS adipocytes, we observed a higher dependency on FAs, and a lower dependency on glucose compared to the controls (Fig. 5g, h). Similar findings were observed in iBAs (Fig. 5i, j). These results suggest that FAs are preferentially utilized in mitochondria in *MTCH2*-deficient human and mouse adipocytes. FAs, derived from the breakdown of triglycerides and other lipids,

undergo oxidation through a series of sequential reactions known as β-oxidation. Therefore, we proceeded to measure the rate of FAO, a pivotal catabolic pathway for energy production, by quantifying the conversion of [9,10-³H(N)]-palmitic acid to ³H₂O. Consistent with the observed decreased FFAs levels, we noted an increase in FAO rate in MTCH2-ablated hMADS adipocytes (Fig. 5k) and iBAs (Fig. 5l). Taken together, these data demonstrate that the loss of MTCH2 shifts substrate utilization towards FAs and increases FAO in both mature human and mouse adipocytes.

## MTCH2 deficiency increases CPT1 activity in adipocytes

To understand the molecular changes that underwrite enhanced mitochondrial respiration upon MTCH2 knockdown, we performed unbiased metabolomics profiling in hMADs adipocytes. Glucose and its glycolysis metabolite, pyruvate, were unaltered by *MTCH2* knockdown (Supplementary Fig. 5a) while lactate and alanine production tended to increase over time, but no significant difference was observed (Supplementary Fig. 5b). Other TCA-cycle metabolites (Supplementary Fig. 5c, d), non-glycolytic glucose metabolites (Supplementary Fig. 5e), glutamate, glutamine, glutathione (Supplementary Fig. 5f), choline and phosphoethanolamine levels were mostly

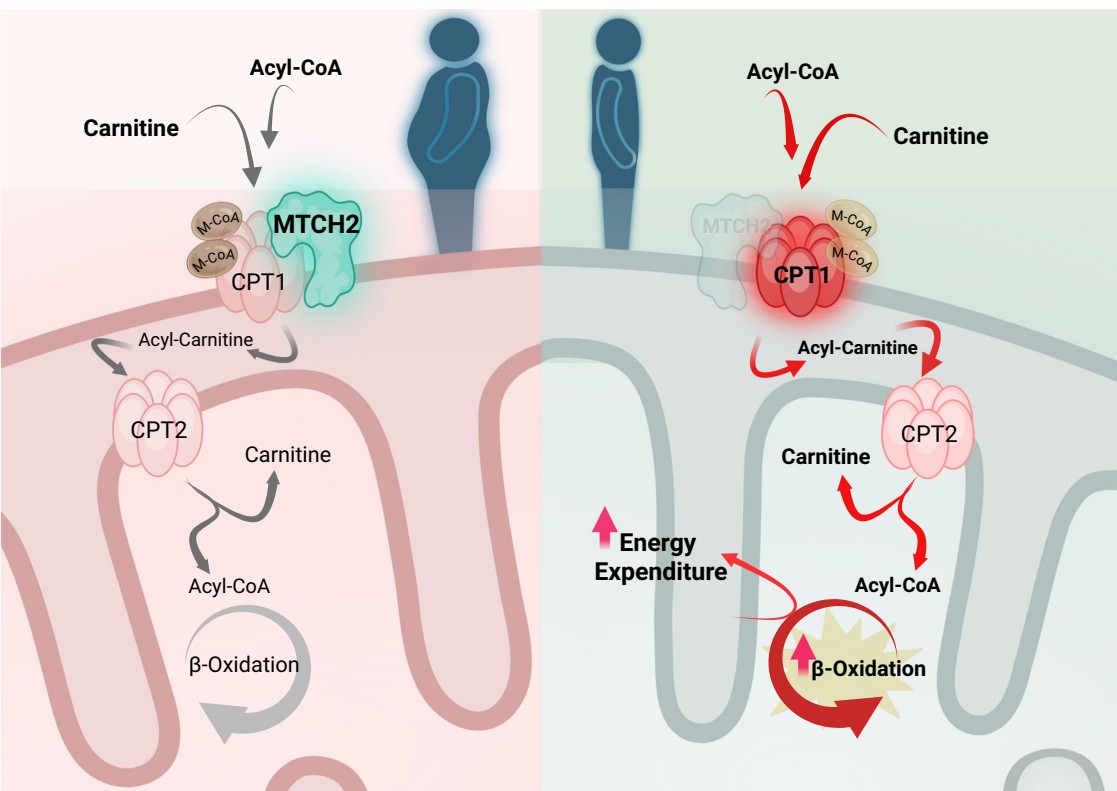

**Fig. 7 | A schematic summary of mitochondrial carrier homolog 2 (MTCH2)-mediated effects on lipid metabolism and energy expenditure (created with BioRender, https://BioRender.com/xew2csf).** MTCH2 expression is positively correlated with obesity in human adipose tissues. MTCH2 interacts with carnitine palmitoyltransferase 1 (CPT1) to regulate CPT1 activity and lipid metabolism in adipocytes. Upon *MTCH2* depletion (indicated by the light green background in the right panel), CPT1 becomes less sensitive to malonyl-CoA (M-CoA), resulting in increased fatty acid oxidation (FAO) in adipocytes and contributing to enhanced energy expenditure in adipose tissue. Red arrows on the right side indicate elevated CPT1 activity and FAO rates.

unperturbed with a slight decreased in phosphocholine levels in *MTCH2* knockdown cells at 6 h time point (Supplementary Fig. 5g). Similarly, NADP⁺, NADPH levels, and NADP⁺/NADPH ratios (Supplementary Fig. 5h) remained unchanged with a trend towards increased NAD⁺/NADH ratios (Supplementary Fig. 5i). Even amino acids, which are known to modulate energy homeostasis in adipocytes[42], were unperturbed by *MTCH2* knockdown (Supplementary Figs. 5b, f and 6a–c). Creatine levels were increased in the *MTCH2* knockdown group at the 3 h time point, while creatine phosphate levels remained unaltered (Supplementary Fig. 6d). We observed no changes in creatine levels (Supplementary Fig. 6e) in iBAT between *Mtch2^{fl/fl}Adip-CreERT2* and control littermates.

MTCH2 has also been reported to regulate membrane lipids and storage lipids levels to meet energy demands[43]. Therefore, we investigated the effects of MTCH2 on various lipid classes. Most major lipid levels remained unchanged; however, phosphatidylserine (PS) levels were increased (Supplementary Fig. 6f, g). Lipid composition analysis revealed a decrease in phosphatidic acid (PA) and storage lipid cholesteryl esters (CE) (Supplementary Fig. 6h, i). Overall, MTCH2 induced mild alterations in membrane and storage lipid levels.

Given the fact that neither mitochondrial protein nor metabolite concentrations were changed, we considered the insertase[25] activity of MTCH2 and resorted to performing mitochondrial proteomics analysis to identify potential client proteins that could mediate FAO in iBAs. However, most mitochondrial protein levels were not altered by *Mtch2* ablation (Fig. 6a), including the proteins involved in lipid metabolism such as CPT1a, CPT1b, AGPAT5, and GPAM (Supplementary Fig. 7a). Intriguingly, CPT1 activity is not only modulated by its availability but is also efficiently controlled by other modulatory factors. Malonyl-CoA acts as an allosteric inhibitor of CPT1 enzymes, crucial for the

regulation of mitochondrial FAO[44]. We therefore assessed the CPT1 activity in isolated mitochondria fractions by quantifying the conversion of L-[methyl-³H]-carnitine and palmitoyl-CoA into palmitoyl-L-[methyl-³H]-carnitine in iBAs, iBAT, and iWAT. Interestingly, we found that CPT1 demonstrated less sensitivity to malonyl-CoA inhibition in adipocytes after *Mtch2* knockdown (Fig. 6b, c). This finding was also validated in mitochondria isolated from *Mtch2^{fl/fl}Adip-CreERT2* mice iBAT (Fig. 6d) and iWAT (Fig. 6e). Importantly, the expression levels of CPT1 protein were not altered in response to *Mtch2* silencing (Supplementary Fig. 7a, b), ruling out changes in CPT1 abundance as an explanation for the altered CPT1 activity. The malonyl-CoA levels were unaltered in iWAT and iBAT in *Mtch2^{fl/fl}Adip-CreERT2* mice (Fig. 6f). Taken together, these results demonstrate that ablation of MTCH2 accelerates FAO through modulating CPT1 activity to meet cellular energy demands.

## MTCH2-CPT1 interacts in the outer mitochondrial membrane

Given MTCH2's localization and function at the mitochondrial outer membrane, it is plausible that MTCH2 influences CPT1 activity through direct protein-protein interactions. Therefore, we further investigated whether MTCH2 interacts with CPT1. We observed a direct interaction between MTCH2 and both CPT1A and CPT1B via proximity ligation assay in iBAs (Fig. 6g, h). This interaction was further validated through immunoprecipitation assays in HEK293T cells overexpressing FLAG-MTCH2, utilizing a FLAG antibody (Fig. 6i). These observed results strongly suggest a physical interaction between MTCH2 and CPT1, which likely influences CPT1 structure and its sensitivity to malonyl-CoA. To further investigate which domain of MTCH2 is involved in its interaction with CPT1, we generated a siRNA-resistant FLAG-MTCH2 wild-type (WT) construct using a lentiviral system. Based on published

structural information[20,45], we produced seven truncation mutants (M1–M7) on the WT backbone (Supplementary Fig. 7c). All constructs were overexpressed in HEK293T cells to assess MTCH2-CPT1 interaction, and were subsequently introduced into iBAs via viral infection, followed by siRNA-mediated knockdown of endogenous MTCH2 to evaluate their effects on CPT1 activity. All mutants were successfully expressed and localized to mitochondria in HEK293T cells, except for the N-terminal truncation mutant (M1) (Supplementary Fig. 7d), suggesting that the N-terminus is critical for proper protein synthesis or stability. Notably, MTCH2-CPT1 interaction was preserved in all other mutants (M2–M6) except for M7 (Supplementary Fig. 7e). Moreover, only M7 construct was able to restore the CPT1 activity induced by MTCH2 depletion (Supplementary Fig. 7f). In addition, we performed bioinformatic analyses to identify other mitochondrial proteins with sequence similarity to the C-terminus of MTCH2, aiming to uncover additional candidates that might interact with CPT1. Among the hits, MTCH1 showed the highest similarity, with 50% sequence identity. However, our previous experiments testing the effects of MTCH1 on cellular respiration in human adipocytes showed no significant impact (Supplementary Fig. 7g), suggesting that MTCH2 interacts with CPT1 through distinct and functionally specific regions not shared with MTCH1, or potentially due to differences in their three-dimensional protein structures. These findings suggest that the C-terminus of MTCH2 is essential for both its interaction with CPT1 and the regulation of CPT1 activity, thereby regulating FAO in adipocytes and contributing to systemic energy homeostasis (Fig. 7).

## Discussion

Increasing mitochondrial oxidative capacity and enhancing energy expenditure are promising strategies to combat obesity. While there are several known targets among the list of positive modulators for mitochondrial oxidative capacity (e.g., AMPK, PGC-1α, etc.)[46] and energy expenditure (e.g., UCP1, ANTs, futile cycles, etc.)[11–13,47], the list of targetable negative modulators is slim. Here, we elucidate the metabolic relevance of MTCH2 in AT, highlighting its crucial role in mitochondrial oxidative function and energy metabolism regulation independent of UCP1. MTCH2 was initially characterized as a regulator of OXPHOS in stem[24] and muscle cells[31], contributing to mitochondrial function. Interestingly, we did not observe any effect of MTCH2 on cell survival and OXPHOS complex protein levels in either human or mouse adipocytes. This suggests its regulation of apoptosis and OXPHOS may be cell type specific and highlights the intricate nature of its regulatory functions across different cellular environments. This molecular role differs from findings in a recent study that employed a constitutive genetic knockout model[48], in which MTCH2 was absent from early development. These differences may be attributed to variations in tissue development and the timing of gene deletion, as in our study where MTCH2 was specifically ablated in AT using a tamoxifen-inducible model. Similarly, knockdown experiments in this study, which demonstrated the dependency of MTCH2 function on CPT1, were performed in both mature human and mouse adipocytes, emphasizing the context-specific role of MTCH2 in AT.

In this study, we found that MTCH2 deletion leads to increased basal respiration, cAMP- and isoproterenol-stimulated uncoupled respiration in human and mouse adipocytes, independent of UCP1. This suggests that the absence of MTCH2 drives a metabolic shift towards heightened mitochondrial activity, uncoupled from ATP production, potentially increasing energy expenditure. Additionally, recent findings show that MTCH2 deletion elevates ATP demand, supported by increased mitochondrial oxidation and nutrient catabolism in HeLa cells[43]. This elevated energy demand also influences FA utilization and alters the lipid profile, leading to a reduction in key membrane lipids, such as phosphatidylethanolamine, phosphatidylserine, cardiolipin, and cholesterol[43]. These lipids are essential for shaping the curvature and determining the rigidity or fluidity of the

inner mitochondrial membrane[49]. In mature hMADs adipocytes, only PS absolute levels were significantly increased. Lipid composition analysis revealed a significant decrease in PA and CE levels. These findings differ slightly from those observed in HeLa cells, which may be due to cell type-specific effects. MTCH2 has been proposed as a phospholipid scramblase[26,50], maintaining mitochondrial membrane integrity by shuffling lipids between bilayer leaflets[51]. These findings suggest that the altered lipid composition and membrane integrity in MTCH2-deficient cells may contribute to the observed changes in mitochondrial respiration.

Mitochondrial dynamics are crucial for maintaining mitochondrial morphology and function, and involve the interplay between mitochondrial fission and fusion processes, regulated by proteins like MFN1, MFN2, OPA1, and DRP1. MTCH2 has been revealed as a selective effector of starvation-induced mitochondrial hyperfusion[23], and shown to cooperate with MFN2 and lysophosphatidic acid synthesis to sustain mitochondrial fusion[21]. In our study, we observed no changes in the abovementioned mitochondrial dynamics-related proteins in human and mouse adipocytes, which is further supported by our electron microscopy images and mitochondrial proteomics data. However, we observed an increase in fission 1 (FIS1) protein in our mitochondrial proteomics with *Mtch2* ablation, which is a specific adapter protein facilitating DRP1 recruitment to mitochondria[52]. Despite the unchanged DRP1 and its phosphorylation levels, elevated FIS1 levels suggest regulation by MTCH2's insertase or yet unknown functions. These data suggest that MTCH2 regulates mitochondrial metabolism independent of mitochondrial dynamics in human adipocytes.

Several previous studies have emphasized the role of MTCH2 in lipid homeostasis[23,28–31]. Overexpression or depletion of MTCH2 can increase or deplete lipid accumulation, respectively, in various models[23,28,29,32], though specific studies in mature AT are lacking. In our study, we found that *MTCH2* silencing leads to reduced levels of FFAs by increasing the rate of FAO in both human and mouse adipocytes. This shift towards increased lipid utilization of mitochondria in adipocytes contrasts with its effects observed in muscle[31]. However, the underlying mechanism for this discrepancy remains unclear. MTCH2-deficient HeLa cells rely on FAO for cellular respiration and proliferation[43], providing additional evidence for the role of MTCH2 in modulating FAO. Furthermore, MTCH2 deletion in HeLa cells leads to nutrient depletion, resulting in alterations in several metabolic pathways, including glyoxylate and dicarboxylate metabolism, TCA cycle, pentose-phosphate pathway, amino acid, nucleotide (purine and pyrimidine), and nitrogen metabolism pathways[43]. Interestingly, in our study, we did not observe significant changes in major metabolites involved in glucose metabolism and TCA cycle, except for creatine. Creatine, an amino acid derivative, is involved in the creatine futile cycle, contributing to systemic energy expenditure in mice[12,53], although its effect in humans is still under debate[54,55]. In our study, we observed increased creatine levels only at very specific time points, suggesting that the metabolic effects of MTCH2 on energy expenditure in adipocytes might be independent of creatine levels.

The import of FAs into the mitochondrial matrix is a critical step that regulates FAO rate, which is often considered a bottleneck. This process involves a series of enzymes and transporters, with CPT1 being the rate-limiting enzyme[14]. Three isoforms CPT1A, CPT1B, and CPT1C have been reported[14,56]. While CPT1A and CPT1B are both expressed in adipocytes and located in the outer mitochondrial membrane, CPT1C is present mainly in neurons and located in the endoplasmic reticulum membrane[56]. Both CPT1A and CPT1B serve as mitochondrial transferases, facilitating the transport of the acyl group from acyl-coenzyme A to carnitine, forming acyl carnitines[56]. In our study, we observed no changes in CPT1 protein abundance in mitochondria, but elevated CPT1 activity in both in vitro and in vivo with MTCH2 ablation, which subsequently contributes to enhanced energy expenditure.

The activity of CPT1 can be regulated by several factors, including malonyl-CoA levels, substrate availability[56], membrane lipid architecture, and mitochondrial shape[57]. In our study, we observed no changes in malonyl-CoA levels in iBAT and iWAT of *Mtch2^fl/fl^Adip-CreERT2* mice, suggesting that the observed elevation in CPT1 activity is not attributable to decreased malonyl-CoA inhibition in AT. It has also been shown that fragmented mitochondria can modulate CPT1 activity[57]. However, our study found no effects of MTCH2 on mitochondrial dynamics in human adipocytes, indicating a distinct function of MTCH2 in adipocytes. Its regulation of mitochondrial metabolism is independent of mitochondrial dynamics in human adipocytes. Crucially, we identified a physical interaction between MTCH2 and CPT1. Truncation mutant analysis further revealed that the C-terminal region of MTCH2 is essential for its interaction with CPT1. This interaction may alter CPT1 conformation or stability within the mitochondrial membrane, thereby influencing its catalytic activity and sensitivity to malonyl-CoA, as observed in this study. MTCH2 has been reported to interact with several key proteins involved in FA synthesis, oxidation, and storage, including CPT1A[23]. Therefore, the physical interaction between MTCH2 and CPT1 implies a direct regulatory mechanism by which MTCH2 may modulate CPT1 activity. Nevertheless, our work sets a stage for a detailed biochemical exploration of MTCH2-CPT1 interaction and how this interaction affects adipose (patho)physiology.

## Methods

The clinical study for human adipose tissues transcriptome was approved by the Ethics Committee of the Hospital District of Southwest Finland and conducted according to the principles of the Declaration of Helsinki. The LOBB study received approval from the Ethics Committee of the University of Leipzig (approval no: 159-12-21052012), and acquisition was performed in accordance with the Declaration of Helsinki. The ACTIBATE study received approval from the Ethics Committee on Human Research of the University of Granada (no. 924) and the Servicio Andaluz de Salud (Centro de Granada, CEI-Granada, Spain). All study participants provided written consent prior to entering the study. All animal experiments in our study were approved by the Veterinary Office of the Canton of Zürich, Switzerland (ZH114/2024).

### Clinical transcriptome study

The clinical study for transcriptome was approved by the Ethics Committee of the Hospital District of Southwest Finland and conducted according to the principles of the Declaration of Helsinki. All subjects (age $40.2 \pm 9.4$ years, BMI $22.8 \pm 2.2$ kg/m$^2$, 20 females and 7 males) provided written consent prior to entering the study[58]. The subjects were screened for medical history and status, and only healthy volunteers were enrolled in the study. PET-CT scan and tissue biopsies were performed[58]. Briefly, the subjects underwent a PET-CT examination after an overnight fast. On the cold exposure day, the subjects spent 2 h wearing light clothing in a room with an ambient temperature of $17 \pm 1$ °C before moving into the PET-CT room, which had an air temperature of 23 °C. During the PET-CT session, one foot of the subject was placed intermittently (5 min in/5 min out) in cold water at a temperature of $8 \pm 1$ °C. Detailed description of PET-CT examinations is available in previous publication[58]. The site of the biopsy was selected based on the cold exposure $^{18}$FDG-PET-CT image that showed activated BAT. A subcutaneous WAT sample was collected from the same incision. The biopsies were obtained under local lidocaine anesthesia by a plastic surgeon at normal room temperature (20 °C) 1 week after the PET-CT examination. Immediately after removal, the tissue samples were snap-frozen in liquid nitrogen and stored at −80 °C until further processing. The RNA from adipose tissue was isolated using the RNeasy Lipid Tissue Mini Kit (QIAGEN), according to the manufacturer's protocol, including the DNase treatment step. Sex

and/or gender of participants was determined based on self-report. No sex- and gender-based analyses have been performed.

### Clinical study of the Leipzig Obesity BioBank (LOBB)

The LOBB is a collection of human body fluids, adipose tissue samples, and associated data designed to expand the knowledge of obesity and related diseases[59,60]. The LOBB covers paired human samples of omental visWAT and abdominal scWAT for three different cohorts and has been performed in accordance with the Declaration of Helsinki. All study participants provided written consent prior to entering the study. The cross-sectional cohort (CSC) and metabolically healthy versus unhealthy obesity cohorts[39] were analyzed in our study. The samples have been collected during elective laparoscopic abdominal surgery, and laboratory measurements of metabolic parameters and body composition were obtained[60]. The CSC comprises 1479 individuals categorized as either normal/overweight (n = 31; 52% women; age: $55.8 \pm 13.4$ years old; BMI: $25.7 \pm 2.7$ kg/m$^2$) or obese (n = 1448; 71% women; age: $46.9 \pm 11.7$ years old; BMI: $49.2 \pm 8.3$ kg/m$^2$). Our analysis utilized RNA-sequencing data from 635 individuals for scWAT and 721 individuals for visWAT, all of which successfully passed quality control (QC).

The cohort for distinguishing metabolically healthy versus unhealthy obesity (MHUO) comprises 29 individuals classified as insulin sensitive (IS; 80% female; age: $46.0 \pm 7.1$ years old; BMI: $46.0 \pm 7.1$ kg/m$^2$; FPG: $5.2 \pm 0.2$ mmol/L; FPI: $27.6 \pm 13.8$ pmol/L) and 39 individuals classified as insulin resistant (IR; 74.4% female; age: $47 \pm 7.6$ years old; BMI: $46.6 \pm 7.1$ kg/m$^2$; FPG: $5.7 \pm 0.3$ mmol/L; FPI: $106.4 \pm 31.6$ pmol/L)[39,60]. All bulk RNA sequencing libraries were sequenced on a Novaseq 6000 instrument at the Functional Genomics Center Zurich (FGCZ). Approval for all studies was obtained from the Ethics Committee of the University of Leipzig (approval no: 159-12-21052012) before the study, and acquisition was performed in accordance with the Declaration of Helsinki. Sex and/or gender of participants was determined based on self-report. No sex- and gender-based analyses have been performed.

### Clinical study of the ACTIBATE study

The study received approval from the Ethics Committee on Human Research of the University of Granada (no. 924) and the Servicio Andaluz de Salud (Centro de Granada, CEI-Granada, Spain)[54,61]. All study participants provided written consent prior to entering the study. A total of 145 young sedentary adults aged 18–25 years old participated in the Activating Brown Adipose Tissue Through Exercise (ACTIBATE) study (ClinicalTrials.gov ID: NCT02365129). Eligible participants were randomly assigned into three groups: a non-exercise group or one of two exercise groups (moderate-intensity exercise and vigorous-intensity exercise). The BRB-seq libraries were prepared at Alithea Genomics Laboratory in Epalinges, Switzerland, according to the following protocols. For the MERCURIUS BRB-seq library preparation service, RNA samples were shipped on dry ice and underwent quantification and quality control before reverse transcription (RT). Depending on the number of samples, the RT was carried out in batches of either 96 or 384, utilizing a set of barcoded oligo-dT primers from BRB-seq kits (96-sample BRB-seq kit, PN10813; 384-sample BRB-seq kit, PN11013; Alithea Genomics) accordingly. The cDNA was pooled into a single tube, column-purified, and subjected to ExoI treatment before proceeding with cDNA second-strand synthesis. The Illumina-compatible Unique Dual Indexed (UDI) library was prepared, and the libraries were sequenced on the Illumina NovaSeq instrument[62]. The FASTQ files were then sample-demultiplexed and aligned to the reference genome using STARsolo version 2.7.9a. The resulting read count matrices were subsequently used for downstream gene expression analysis. Our analysis was conducted using data from 60 individuals (80% women; age: $22.1 \pm 2.1$ years old; BMI: $25.5 \pm 4.4$ kg/m$^2$) across these groups, all of whom had complete information on *MTCH2*

expression, BMI, fat mass, and relative oxygen consumption ($VO_2$) at the maximum effort test levels prior to exercise. Among these 60 individuals, 30 also had data on meal-induced thermogenesis, which was assessed as the area under the curve (AUC) of the entire post-prandial response relative to meal energy intake (expressed as a percentage). Sex and/or gender of participants was determined based on self-report. No sex- and gender-based analyses have been performed.

## Mouse experiments

The $Mtch2^{fl/fl}$ mouse strain was kindly provided by Prof. Atan Gross Lab (Weizmann Institute of Science, Israel) in 129/SVJ l background. $Adip$-CreERT2 mice were generated by Dr. Evan Rosen lab (Beth Israel Deaconess Medical Center and Harvard Medical School) in C57BL/6J background. Inducible adipose tissue-specific ablation of MTCH2 was achieved by crossing $Mtch2^{fl/fl}$ mice with $Adip$-CreERT2 mice[63]. All animal experiments in our study were approved by the veterinary office of the Canton of Zürich. All male mice were used in this study. They were housed 2–5 littermates per cage in ventilated cages at standard housing conditions (22 °C, 40% humidity, 12 h reversed light and dark cycle, the dark phase starting at 7 am), with ad libitum access to chow diet (18% proteins, 4.5% fibers, 4.5% fat, 6.3% ashes, Provimi Kliba SA) and water. The health status of all mice was regularly monitored according to the Federation of European Laboratory Animal Science Associations guidelines. For primary brown adipocytes isolation, 6-week-old male mice were used. For other animal experiments, 7–9 weeks old male mice were orally administered tamoxifen (2 mg/mouse in sunflower oil, Sigma-Aldrich) to induce recombination of the floxed allele for further study on chow diet or HFD. The exact numbers for mice experiments were indicated in each figure legend.

## Cell culture and transfection

hMADS cells derived from the prepubic fat pad of a 4-month-old male were kindly provided by Dr. Amri and cultured following established protocols[64]. Briefly, cells (passage 6 to 10) were cultured in low-glucose DMEM supplemented with 15 mM HEPES, 10% FBS, 2 mM L-glutamine, 1% Penicillin/Streptomycin, and 2.5 ng/ml recombinant human FGF-2 (Peprotech) in a normoxic humidified cell culture incubator at 37 °C with 5% $CO_2$. The medium was refreshed every other day, and FGF-2 was withdrawn upon reaching confluence. Differentiation was induced in cells 48 h post-confluent (day 0) using DMEM/Ham's F12 media (Lonza) supplemented with 10 µg/ml transferrin, 10 nM insulin, 0.2 nM triiodothyronine, 1 µM dexamethasone, and 500 µM isobutyl methyl-xanthine (IBMX). From days 2 to 9, cells were maintained in adipogenic medium supplemented with 100 nM rosiglitazone. All chemicals were procured from Sigma-Aldrich, except for rosiglitazone (Adipogen). Cells were cultured until day 18 in the absence of rosiglitazone to facilitate the maturation of white adipocytes. To generate beige adipocytes, cells received additional rosiglitazone pulses between days 14 and 18. The knockdown of candidate genes was achieved by delivering 50 nM siRNA pools (Microsynth) into mature adipocytes on day 15 using Lipofectamine RNAiMAX (Invitrogen), following the manufacturer's instructions. A control siRNA pool was used in all experiments as a control. Following a 24-h transfection period, the medium was replaced with fresh adipogenic medium containing 100 nM rosiglitazone. Adipocytes were then cultured until day 18, at which point cellular respiration was assessed, or cells were harvested for further analysis. siRNA sequences were shown in Supplementary Table 6.

Immortalized murine brown adipocytes (iBAs) were kindly provided by Prof. Klein[65]. They were preadipocytes, derived from the iBAT stromal-vascular fraction of late fetal and newborn C57Bl/6 mice (both genders) and immortalized through SV40 antigen introduction. These preadipocytes (passage 2–6) were cultured on collagen-coated plates in DMEM supplemented with 10% FBS and 1% Pen/Strep (Gibco) in a normoxic humidified cell culture incubator at 37 °C with 5% $CO_2$. Upon reaching confluence, adipogenic differentiation was initiated by adding IBMX (500 µM), dexamethasone (1 µM), insulin (20 nM), T3 (1 nM), and indomethacin (125 µM) to the medium. All chemicals were sourced from Sigma-Aldrich. Following 48 h, the medium was replaced with fresh medium containing insulin and T3, with subsequent medium changes every other day. For siRNA-mediated gene knockdown, differentiating adipocytes were trypsinized on day 5, counted, and replated on collagen-coated multi-well plates to lower cell density. Subsequently, cells were transfected with target siRNA pools (100 nM) on day 6. On day 7, cells were replenished with fresh medium containing insulin and T3. On day 9, cells were harvested for cellular respiration measurements or further analysis.

HEK293T cells were obtained from Abcam (ab255449) and were cultured in DMEM with 10% FBS and 1% penicillin/streptomycin. For transient overexpression of mouse FLAG-MTCH2, cells were transfected with plasmids using Lipofectamine 2000 when they reached 70–80% confluence. After a 12-h incubation period, fresh medium was replaced, and cells were harvested for further analysis after 48 h. All cell lines utilized in this study were routinely subjected to testing for mycoplasma contamination, and all results were consistently negative throughout the entire duration of the study.

## Lentivirus packaging

HEK293-LTV cell lines were purchased from Cell Biolabs Inc. (LTV-100). For lentivirus packaging, the target plasmids were transfected into HEK293T LTV cells together with pMD2.G (Addgene, 12259) and psPAX2 (Addgene, 12260) by polyethylenimine in OptiMEM medium. The virus-containing medium was collected and concentrated in PEG-it Virus Precipitation Solution (SBI, LV825A-1) according to the manufacturer's instructions.

## Isolation and differentiation of the stromal-vascular fraction-derived primary brown adipocytes

The whole iBAT depots from $Mtch2^{fl/fl}$ and of $Mtch2^{fl/fl}Adip$-CreERT2 (6 week old) mice were minced with scissors and incubated in a collagenase buffer (25 mM $NaHCO_3$, 12 mM $KH_2PO_4$, 4.8 mM KCl, 1.2 mM $MgSO_4$, 120 mM NaCl, 5 mM glucose, 1.4 mM $CaCl_2$, 2.5% BSA, 1% penicillin–streptomycin, pH = 7.4, 1 mg/mL) collagenase (C6885-1G, Sigma-Aldrich) for 30 min under agitation. An equal volume of culture medium (DMEM medium (61965026, Gibco) containing 10% FBS and 1% penicillin–streptomycin) was added and centrifuged for 5 min at 300 x g. The pellet (SVF cells) was resuspended in a 10 mL culture medium and passed through 100-mm cell strainers. SVF cells were then plated on collagen-coated plates. After the cells reached full confluence, the differentiation was induced by adding IBMX (500 µM), dexamethasone (1 µM), insulin (5 µg/mL), and rosiglitazone (1 µM) to the medium. After 48 h, the induction medium was replaced by a maintenance cocktail (complete DMEM supplemented with 1 mg/mL insulin), which was refreshed every other day. On day 5, adipocytes were trypsinized, counted, and replated with $Ucp1$ targeting siRNA pools (100 nM) on collagen-coated 96-well Seahorse microplates for cellular respiration assay. Subsequently, cells were treated with 1 µM 4-hydroxytamoxifen for 3 days to induce $Mtch2$ KD. On day 9, cells were harvested for cellular respiration measurements.

## RNA extraction, cDNA synthesis, and quantitative RT-PCR

Total RNA was extracted from cells or tissues using Trizol reagent (Invitrogen) following the manufacturer's instructions. DNase treatment (NEB BioLabs) was included to eliminate traces of genomic DNA. Reverse transcription was conducted to generate a cDNA library using the High-Capacity cDNA Reverse transcription kit (Applied Biosystems). Quantitative PCR was carried out on a ViiA7 system (Applied Biosystems), analyzed by ViiA7 Ruo v1.2.3. Relative mRNA levels were normalized to the expression of RPL13A1 (human fat samples) or TBP (cell culture) and were calculated by the ΔΔCt method. Primer sequences are found in Supplementary Table 7.

## RNA sequencing, mapping, and analysis

Extracted RNA was quality-checked using Tapestation (Agilent). All samples had a RIN value greater than 8. For library preparation, the TruSeq mRNA sample preparation kit (Illumina) was used. Sequencing was conducted as 50 bp single reads and 7 bases index reads on an Illumina HiSeq2000 instrument. Approximately 20–30 million reads per sample were obtained. RNA sequencing data were processed using kallisto with the Gencode human reference genome release 29. The different datasets were processed using the same pipeline and reference genome to ensure comparability. The quality of the fastq files was assessed using the FastQC program (Babraham Bioinformatics, Babraham Institute, 2010). The raw gene counts obtained from kallisto were further processed by DESeq2 to identify the DE genes. The PCA plots after variance-stabilizing transformation normalization of the top 500 most variable genes were visually inspected to identify and remove any outlier sample, particularly in the case of supraclavicular BAT. DE genes with a false discovery rate (FDR) cut-off <0.05 were selected, and the overlapping DE genes in human adipose tissues, hMADS, and murine adipocytes were further processed.

## Seahorse assays

Cellular respirations were monitored by XF96 Extracellular Flux Analyzer (Agilent Seahorse). For the measurement of cellular respiration, cells were differentiated on collagen-coated 96-well Seahorse microplates. On the day of experiment, the adipogenic medium was replaced with XF assay medium (pH 7.4, Seahorse Bioscience) supplemented with glucose (4.5 g/L for iBAs and 1 g/L for hMADs; Sigma-Aldrich), 2 mM sodium pyruvate (Invitrogen), and 2 mM L-glutamine (Invitrogen). The oxygen consumption rate (OCR) was measured using the Extracellular flux analyzer XF96 (Agilent). Test compounds were injected to obtain the following concentrations: 1 µg/ml Oligomycin, 1 µM isoproterenol for iBAs, 0.5 mM dibutyryl cAMP for hMADS, 1 µg/ml FCCP, 3 µM rotenone with 2 µg/ml antimycin A. All compounds were purchased from Sigma-Aldrich, except for Oligomycin (Adipogen). Non-mitochondrial respiration was subtracted to obtain basal, basal uncoupled, stimulated uncoupled, and maximal mitochondrial respiration. Cell respiratory analysis was done by Wave 2.6.0. (Agilent Seahorse).

For the fuel dependence assay, cells were differentiated on collagen-coated 96-well Seahorse microplates. On the day of the experiment, the adipogenic medium was replaced with XF assay medium (pH 7.4, Seahorse Bioscience) supplemented with 10 mM glucose (Sigma-Aldrich), 1 mM sodium pyruvate (Invitrogen), and 2 mM L-glutamine (Invitrogen). The final concentrations for compound injections were: 100 µM etomoxir (Sigma-Aldrich, an irreversible CPT1 inhibitor preventing the transport of fatty acids into the mitochondrial matrix for further metabolization), 6 µM BPTES (Sigma-Aldrich, a glutaminase inhibitor), and 2 µM UK5099 (Sigma-Aldrich, a pyruvate carrier inhibitor). Fuel dependence was calculated by the difference between the basal OCR and after single inhibitor injection over the difference between the basal OCR and after double inhibitor injection. Cell respiratory analysis was done by Wave 2.6.0. (Agilent Seahorse).

## Protein extraction and immunoblotting

Differentiated adipocytes and adipose tissue samples were homogenized in RIPA buffer (50 mM Tris-HCl, pH 7.4, 150 mM NaCl, 2 mM EDTA, 1.0% Triton X-100, 0.5% sodium deoxycholate) supplemented with protease inhibitors (Complete, Roche) and phosphatase inhibitor cocktail (Thermo Fisher)[66]. The homogenates were then centrifuged at 12,000 × $g$ for 15 min at 4 °C to remove debris and collect the whole cell lysates. The protein concentration of the supernatants was determined by the DC Protein Assay (Bio-Rad) using by SynergyMx plate reader (BioTek). Equal amounts of protein were loaded and separated on 12% SDS-polyacrylamide gels. The proteins were transferred to a nitrocellulose membrane (Bio-Rad) and incubated with specific antibodies: MTCH2 (1:1000, Invitrogen), UCP1 (1:1000, Pierce, Thermo Fisher), phospho-HSL (Ser660; 1:1000, Cell Signaling), total HSL (1:1000, Cell Signaling), OXPHOS (1:1000, Abcam), phospho-DRP (Ser616; 1:1000, Thermo Fisher) phospho-DRP (Ser637; 1:1000, Thermo Fisher), DRP1 (1:1000, Cell Signaling), OPA1 (1:1000, Cell Signaling), MFN1(1:1000, Proteintech), MFN2 (1:1000, Proteintech), CPT1A (1:1000, Proteintech/Cell Signaling), CPT1B (1:1000, Invitrogen/Cell Signaling), HSP90 (1:1000, Cell Signaling), γ-tubulin (1:10.000, Sigma-Aldrich), VDAC (1:1000, Proteintech) and p70S6 kinase (1:1000, Cell Signaling). The signal of the HRP-conjugated secondary antibodies (1:10.000, Cell Signaling) was visualized using the ChemiDoc MP Imaging System (Bio-Rad) and the Image Quant system (GE Healthcare Life Sciences). Image Lab 6 and ImageJ 1 were used to analyze Western blots and images.

## Indirect calorimetry

Indirect calorimetry measurements were conducted using the Promethion system (Sable Systems) following the manufacturer's guidelines. Male *Mtch2*[fl/fl] and *Mtch2*[fl/flxAdip-CreERT2] mice were individually housed in metabolic cages and allowed to acclimate for 48 h prior to metabolic recording. Energy expenditure, $CO_2$ production, and $O_2$ consumption, as well as locomotor activity, food intake, and water intake, were monitored continuously throughout the measurement period. Energy expenditure ANCOVA analysis was performed by R version 4.5.0 and RStudio 2024.

## Intraperitoneal glucose tolerance test

After 12 weeks of HFD, male *Mtch2*[fl/fl] and *Mtch2*[fl/flxAdip-CreERT2] mice were subjected to a 6-h fasting to assess glucose tolerance, during which the experimental animals were transferred to a clean cage without access to food. Prior to the glucose challenge, mice were weighed, and fasting glucose levels were measured via a small tail clip using a standard glucometer (ACCU-CHEK Aviva, Roche), D-glucose (Sigma-Aldrich) was then injected intraperitoneally at dose 1 mg/g body weight. Blood glucose levels were subsequently measured at 15, 30, 60, 90, and 120 min after glucose injection using the glucometer.

## Body composition measurement

The body composition of live mice was measured using a magnetic resonance imaging technique (EchoMRI 130, Echo Medical Systems). Analysis of fat and lean mass was performed using EchoMRI 14 software.

## Tissue harvest

Animals were euthanized individually in a carbon dioxide atmosphere. Following euthanasia, all tissues were dissected and promptly snap-frozen in liquid nitrogen for subsequent processing. Popliteal lymph nodes were carefully removed from the scWAT for both gene and protein expression analyses. For the isolation of RNA and proteins, the entire adipose tissue depot was homogenized.

## Lipolysis measurement

Lipolysis was assessed in differentiated hMADs and iBAs. Cells were washed with pre-warmed PBS and then starved for 2 h in phenol red-free low-glucose medium (Gibco) supplemented with 1% free acid-free BSA. For stimulated lipolysis, cells were treated with 0.5 mM dibutyryl cAMP for hMADs or 1 µM isoproterenol for iBAs for an additional 1 h at 37 °C in a humidified $CO_2$ incubator. Media were collected for glycerol analysis using the Glycerol Reagent (Sigma-Aldrich) and non-esterified fatty acid analysis using the Wako NEFA kit, according to the manufacturer's instructions. Both assays were normalized to protein content. Absorbance at the indicated wavelengths was measured using the microplate reader Synergy MX and Gen5 software (BioTek).

## Malonyl-CoA, triglyceride, and cholesterol levels

The levels of malonyl-CoA were measured by an ELISA kit (CSB-E12896m; Cusabio, Houston, TX, USA) following the manufacturer's instructions (CUSABIO). Plasma triglyceride and cholesterol levels were measured by triglyceride reagent (Sigma-Aldrich) and cholesterol assay kit (LabAssay ™ Cholesterol, FUJIFILM), respectively.

## Cell viability tests

The cell viabilities were measured in hMADS and iBAs with control siRNA or MTCH2 siRNA transfection at different time points using the Cell Counting Kit-8 reagents (Abcam) following the manufacturer's instructions.

## Immunoprecipitation

HEK293T cells overexpressed with FLAG-tagged MTCH2 were homogenized in RIPA buffer (50 mM Tris-HCl, pH 7.4, 150 mM NaCl, 2 mM EDTA, and 1.0% Triton X-100) supplemented with protease inhibitors (Roche) and phosphatase inhibitors (Thermo Fisher). The homogenates were then centrifuged at $12,000 \times g$ for 15 min at 4 °C to remove debris and collect the whole cell lysates. After three washing steps with cold RIPA buffer, 20 μl of anti-FLAG beads (Sigma-Aldrich) were added, and the mixture was incubated overnight at 4 °C on a rotator. Subsequently, the beads were then washed with cold RIPA buffer six times. The proteins were eluted by boiling at 95 °C for 5 min in 2× Laemmli buffer. The eluted proteins were then analyzed by immunoblotting using FLAG (1:1,000 dilution, Cell Signaling), CPT1a (1:1000, Proetintech), and CPT1b (1:1000, Invitrogen) antibodies.

## Fatty acid oxidation

The fatty acid oxidation rates were assessed in differentiated hMADs and iBAs by quantifying the conversion of [9,10-$^3$H(N)]-palmitic acid to $^3$H$_2$O. In brief, cells were incubated for 3 h in serum-free adipogenic medium containing 50 μM carnitine (Sigma-Aldrich), 50 μM fatty acid-free BSA (Sigma-Aldrich), 100 μM palmitic acid (Sigma-Aldrich), and 1 μCi/mL 9,10-$^3$H palmitic acid (NET043001MC, PerkinElmer, Revvity). The supernatants were transferred into glass vials with rubber stoppers, and the released $^3$H$_2$O was absorbed by filter paper for 48 h at 37 °C. Subsequently, the filter paper was transferred into a scintillation vial containing 5 ml of scintillation fluid for overnight incubation. Finally, the radioactivity was measured using a standard scintillation counter.

## Transmission electron microscopy (TEM) images

Differentiated hMADS adipocytes (day 15) and differentiating iBAs (day 6) were transfected with either control siRNA or MTCH2-targeting siRNA pools. The medium was refreshed the following day. On day 18 for hMADS adipocytes and day 9 for iBAs, cells were fixed in 2.5% glutaraldehyde in PBS to crosslink proteins and preserve cellular structures, followed by post fixation using the OTO method: 2% osmium tetroxide, 1% freshly prepared thiocarbohydrazide, and a second treatment with 1% osmium tetroxide. Samples were then block-stained with 1% uranyl acetate, dehydrated through an ascending ethanol series, and gradually infiltrated with Epon-Araldite resin. Fully infiltrated samples were embedded in fresh resin and polymerized overnight at 60 °C. The polymerized blocks were removed from the molds, trimmed, and sectioned using an ultramicrotome. Ultrathin sections (80–100 nm) were collected on cleaned and glow-discharged silicon wafer chips. The chips were mounted onto SEM stubs using silver paint. Imaging was performed using a TFS Magellan 400L FE-SEM at 1.8 kV and 0.8 nA with backscattered electron detection and a pixel size of 10 nm. Large-area scans were acquired by stitching individual image tiles.

## Mitochondria isolation

Mitochondria were isolated following an established protocol with slight modification[67]. For differentiated adipocytes, cells were washed with ice-cold PBS and then collected in ice-cold isolation buffer (STE; 250 mM sucrose, 5 mM TRIS, 2 mM EGTA, 0.5% fatty acid-free BSA) in the presence of protease inhibitors. Dissected adipose tissues were also washed several times with ice-cold PBS and then cut into small pieces in the ice-cold isolation buffer. The samples were homogenized in the isolation buffer using a Potter-Elvehjem PTFE pestle and glass tube. The resulting homogenate was filtered and centrifuged at $800 \times g$ for 10 min. The supernatant was collected and transferred to a new tube, and centrifuged again at $800 \times g$ for 10 min. The supernatant was transferred and centrifuged at $8000 \times g$ for 10 min to sediment the mitochondrial fraction. Excessive residual fat was removed from the tube walls, and the pellet containing mitochondria was resuspended in fresh STE without BSA. During the final centrifugation step, the initial speed of $2000 \times g$ was increased after 5 min to $4000 \times g$ for another 5 min. The final pellet was resuspended in a minimal amount of STE buffer without BSA, and the protein content was quantified using Bradford reagents (Thermo Fisher).

## Metabolomic analysis

For metabolomic analysis, samples were prepared and accessed according to protocol[68], as described below. Differentiated hMADS adipocytes on day 15 were transfected with control or MTCH2 siRNA pools (Microsynth) using Lipofectamine RNAiMAX (Invitrogen), following the manufacturer's instructions. A control siRNA pool was used as a control. Following a 24-h transfection period, the medium was replaced with fresh adipogenic medium containing 100 nM rosiglitazone. Adipocytes were then treated with cAMP for 1 h, 3 h, and 6 h on day 18, at which point adipocytes were harvested for cellular metabolites measurements between control and MTCH2 KD groups (n = 4 biological replicates per group per time point, 24 samples in total). Differentiated adipocytes were quenched with liquid nitrogen after washing with ice-cold PBS. Intracellular metabolites were then extracted using a pre-chilled mixture (−80 °C) of methanol/chloroform/water, 6.75:0.75:2.5 (v/v/v). The cellular extracts were collected and incubated on dry ice for 30 min. Subsequently, the extracts were centrifuged at 0 °C and $18,000 \times g$ for 15 min. The supernatants were collected and dried using a stream of nitrogen gas. All samples were maintained on dry ice during processing. Media samples were collected at the specific time points and mixed with 2 volumes of LC-grade methanol (Sigma-Aldrich). The dried extracts were reconstituted in 150 mM potassium phosphate buffer (K$_2$HPO$_4$, pH 7.4) in prepared in deuterium oxide (D$_2$O) for NMR analysis. A fixed amount of trimethylsilyl propionate-d$_4$ sodium salt (TSP-d$_4$, Cambridge Isotope Laboratories) was added to each sample as an internal standard (0.05 mM for cell extracts and 0.2 mM for media extracts) to enable quantification. $^1$H-NMR spectra were acquired using a Bruker AVANCE II 600 MHz spectrometer equipped with a 5 mm TCI cryoprobe, employing the noesygppr1d pulse sequence (TopSpin 3.5, Bruker). Metabolites were quantified using Chenomx NMR suite (Chenomx Inc.), with the TSP peak at 0.00 ppm serving as the chemical shift and concentration reference. The absolute concentration of TSP was calibrated using the ERETIC2 method (TopSpin 3.5, Bruker), based on an in-house reference sample of dimethyl sulfone (Sigma-Aldrich) prepared in the same buffer as the test samples. Metabolite concentrations (μM) were normalized to the protein mass (mg) of each extract. Protein content was determined using the Pierce BCA Protein Assay Kit (Thermo Fisher Scientific), following the manufacturer's instructions. NMR concentrations (in μM) were converted to total nmoles per sample using the following formula, accounting for sample volumes and reconstitution steps: $C$ (nmol) = [$C_{NMR}$ concentration (μM) × 0.22 mL × 7 mL]/0.3 mL. Data are presented as mean ± SEM and analyzed using a two-tailed Student's t-test.

## Lipidomics

Differentiated hMADS adipocytes on day 15 were transfected with control or MTCH2 siRNA pools (Microsynth) using Lipofectamine RNAiMAX (Invitrogen), following the manufacturer's instructions. A

control siRNA pool was used as a control. Following a 24-h transfection period, the medium was replaced with fresh adipogenic medium containing 100 nM rosiglitazone. Differentiated adipocytes were harvested on day 18 for lipids measurements between control and *MTCH2* KD groups (n = 6 biological replicates per group, 12 samples in total). Differentiated adipocytes were washed with ice-cold PBS and quenched with liquid nitrogen. Subsequently, adipocytes were scraped with both cold PBS and methanol before being transferred to 2 mL Eppendorf. These samples were further used for lipidomics analysis. Comprehensive targeted lipidomics was accomplished using a flow-injection assay based on lipid class separation by differential mobility spectroscopy (DMS) and selective multiple reaction monitoring (MRM) per lipid species (Lipidyzer platform)[69,70]. After the addition of >60 deuterated internal standards, lipids were extracted using methyl tert-butyl ether. Organic extracts were combined and subsequently dried under a gentle stream of nitrogen and reconstituted in running buffer. Lipids were then analyzed using flow-injection in MRM mode employing a Shimadzu Nexera series HPLC and a Sciex QTrap 5500 mass spectrometer. Each sample was analyzed using MRM in two consecutive flow-injection analysis (FIA) runs with positive and negative polarity switching. In the first run, PC, PE, PG, PI, PS, and SM were separated with the SelexION DMS cell using field asymmetric ion mobility mass spectrometry after ionization in the Turbo V source of the mass spectrometer. 1-propanol was added to the curtain gas as chemical modifier to improve DMS separation. In the second run, FFA, TAG, DAG, CER, DCER, LCER, LPC, LPE, PA, HCER, and CE were measured with the DMS-cell switched off. 75 μL of the reconstituted samples were injected using a Shimadzu SIL 30AC autosampler into the running solvent (10 mM ammonium acetate in dichloromethane: methanol (50:50, vol/vol)) pumped at an isocratic flow rate of 8 μL/min by a Shimadzu Nexera LC30 system. Then the flow rate was ramped to 30 μL/min for 2 min to allow for washing[69]. Flow rate increased after 8.7 min for first run and 9.1 min for the second run. Acquired lipid data were processed using the Shotgun Lipidomics Assistant (SLA) software. SLA oversees three processes in the DMS lipidomics workflow: 1) analysis of DMS tuning data to optimize and update methods, 2) analyzing QC and suitability test data to assess the relative lipid standard response, and 3) analyzing DMS lipidomics data files to generate quantitative lipid profiles[70]. Data are presented as mean ± SEM and analyzed using a two-tailed Student's t-test.

## CPT1 activity

CPT1 activity was determined following methods[71,72] with minor modifications. The activity was assayed in intact isolated mitochondria (10–20 μg of protein), measuring the formation of palmitoyl-L-[methyl-³H] carnitine from L-[methyl-³H] carnitine (1 μCi; Anawa) and palmitoyl-CoA (150 μM). The assay buffer comprised 117 mM Tris-HCl, 0.28 mM reduced glutathione, 4.4 mM ATP, 4.4 mM MgCl2, 16.7 mM KCL, 2.2 mM KCN, 40 mg/L rotenone, 0.5% BSA, and 5 mM L-carnitine. Briefly, the assay was initiated by adding mitochondrial suspension to 80 μl of a reaction buffer containing palmitoyl-CoA and 1 μCi L-[methyl-3H] carnitine, in a total volume of 100 μl. The reaction was conducted at 37 °C for 8 min and then stopped by the addition of 100 μl ice-cold 1 N HCL. For the malonyl-CoA suppression test, the assay was performed in the presence of various malonyl-CoA concentrations (0.1, 1, 10, 20 μM). Palmitoyl-[³H] carnitine was extracted in 400 μl water-saturated butanol through three washes with distilled water and subsequent re-centrifugations at 1000 × g for 5 min to separate the butanol phase. Subsequently, 100 μl of the butanol phase containing radiolabeled palmitoyl-[³H] carnitine was collected and counted in 5 mL scintillation counting solution using a standard scintillation counter.

## Proteomics (LC-MS/MS DIA)

Differentiating iBAs were trypsinized on day 5, counted, and replated on collagen-coated multi-well plates to lower cell density. Subsequently,

cells were transfected with control siRNA and target *Mtch2* siRNA pools (100 nM) on day 6 (n = 4 per group, 8 samples in total, each sample was processed and analyzed once). On day 7, cells were replenished with fresh medium containing insulin and T3. On day 9, adipocytes were harvested for mitochondrial protein isolation and purification[25]. Isolated and purified mitochondrial proteins were prepared by boiling them in a final concentration of 4% SDS for 10 min at 95 °C, followed by mechanical lysis using a tissue homogenizer (2 × 2 min cycles at 30 Hz) and high-intensity focused ultrasound. Proteins were reduced and alkylated by adding Tris(2-carboxyethyl) phosphine and 2-Chloroacetamide to a final concentration of 5 mM and 15 mM, respectively. The samples were then incubated for 30 min at 30 °C with agitation at 700 rpm under light protection. Subsequently, the samples were diluted with pure ethanol to achieve a final concentration of 60% EtOH (v/v). Then, carboxylated magnetic beads (hydrophobic and hydrophilic) were added to the samples, and the proteins were bound to the beads for 30 min at room temperature. The beads were then washed three times with 80% EtOH (KingFisher Flex System, Thermo Fisher Scientific). For enzymatic digestion, the beads were incubated with trypsin in 50 mM TEAB overnight at 37 °C. The remaining peptides were extracted from beads with H₂O, and the two elutions were combined and dried down. The dried samples were resolubilized in 20 μl of MS-solvent (3% acetonitrile, 0.1% formic acid), and 1 μl of indexed retention time peptides (iRT; Biognosys AG) was spiked in each sample for future retention time calibration. Mass spectrometry analysis was performed on an Orbitrap Fusion Lumos (Thermo Scientific) equipped with a Digital PicoView source (New Objective) and coupled to an M-Class UPLC (Waters). The solvent composition at the two channels was 0.1% formic acid for channel A and 0.1% formic acid, 99.9% acetonitrile for channel B. For each sample 2 μL of diluted peptides were loaded on a trap column MZ Symmetry C18 Trap Column (100 Å, 5 μm, 180 μm × 20 mm, Waters), followed by nanoEase MZ C18 HSS T3 Column (100 Å, 1.8 μm, 75 μm × 250 mm, Waters). The peptides were eluted at a flow rate of 300 nL/min by a gradient from 5 to 22% B in 80 min, to 32% B in 10 min, and an immediate increase to 95% B for 10 min. Samples were acquired in a randomized order. The mass spectrometer was operated in data-independent mode (DIA), acquiring a full-scan MS (396–1000 m/z) at a resolution of 60,000 at 200 m/z. Followed by higher-energy collision dissociation fragmentation on 70 windows with 8 m/z width between 400 and 960 m/z. The MS2 resolution was set to 15,000, and a normalized collision energy of 33% was used. The acquired data-independent acquisition spectra were processed with DIA-NN (Version 16) using a library-free approach[73].

## Proximity ligation assay

Differentiated iBAs were fixed with paraformaldehyde for 20 min at room temperature. Subsequently, a proximity ligation assay was performed using the kit from Sigma-Aldrich following the manufacturer's instructions. For the assay, anti-rabbit antibody (MTCH2, Invitrogen) and anti-mouse antibodies (CPT1A from Proetintech, CPT1B from Invitrogen) were used. Additionally, an incubation without antibodies served as the negative control. The images were acquired using an Olympus FLUO-VIEW 3000 confocal microscope and processed using ImageJ.

## Molecular cloning

pCMV-FLAG-MTCH2 expression plasmid was generated by Dr. Salvatore Modica (Vector & Virus Production Platform). The Agilent Quik-Change II Site-Directed Mutagenesis Kit was used to generate an siRNA-resistant construct of MTCH2. To build the lentiviral constructs, FLAG-MTCH2 was inserted into a pLenti-CMV-MCS-BSD vector to obtain the pLenti-FLAG-MTCH2 construct. Using overlapping PCR, various truncated fragments derived from the wild-type MTCH2 were cloned and inserted into the pLenti-FLAG to generate corresponding truncation mutants. All plasmids for the cell culture work were extracted using NucleoBond Xtra-Midi Kit (Macherey-Nagel, product no. 740410.100).

## Statistics and reproducibility

For in vivo studies, mice were randomly assigned to experimental groups. The "n" values represent independent biological replicates for cell experiments or individual mice for in vivo experiments, with specific details provided in each figure legend. Except for the metabolomics, lipodomics, and proteomics experiments, each experiment was repeated independently two to three times. Results are presented as mean ± SEM. Statistical analyses were performed using GraphPad Prism 10. The unpaired two-tailed Student's $t$-test was used for pairwise comparison, two-sided ANCOVA for energy expenditure analysis, and two-sided Pearson correlation for linear regression. The p values were calculated to determine statistical differences and provided in each figure.

## Reporting summary

Further information on research design is available in the Nature Portfolio Reporting Summary linked to this article.

## Data availability

The RNA-seq data used in this study for the human BAT/WAT and pure murine adipocytes transcriptome are available in the European Nucleotide Archive under accession code PRJEB20634. The hMADS RNA-seq data used in this study are available in the European Nucleotide Archive under accession code PRJEB38756. The Proteomics data generated in this study have been deposited in ProteomeXchange under accession code PXD054881. Metabolomics and lipidomics data generated in this study have been deposited in Mendeley Data (https://data.mendeley.com/datasets/cmhzgdr4fb/1). The human RNA-seq data from the LOBB cohort and the ACTIBATE cohort reported in this study cannot be deposited in a public or controlled-access repository (such as dbGaP or EGA) because written informed consent and local data protection regulations prohibit transfer of data to external repositories. Access to the LOBB dataset requires approval by the LOBB steering committee and is only available on request from M. Blüher (matthias.blueher@medizin.uni-leipzig.de). Access to the ACTIBATE dataset is only available on request from C. Wolfrum (christian.wolfrum@ntu.edu.sg). All requests will be reviewed within 1–2 weeks. Data use is restricted to non-commercial research purposes only, and a data use agreement is not required for academic research. All other data generated in this study are provided in the Supplementary Information/Source data file. Source data are provided with this paper.

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

## Acknowledgements

The authors thank Manuel Klug for his valuable input and technical support for animal studies. The authors thank Dr. Salvatore Modica for generating the vectors. The authors also thank Dr. Siew-Liang Foong for his excellent support. Proteomic analysis was performed by the

Functional Genomics Center Zurich. Metabolomics and lipidomics were performed by Leiden University Medical Center (LUMC). C. Wu was supported by the Future Food Initiative Fellowship (RC: 13819). This study was supported and funded by Boehringer Ingelheim (C. Wolfrum). This project also received funding from the European Union's Horizon 2020 research and innovation program under the Marie Skłodowska-Curie grant agreement No 945478 (M. Balaz), the Slovak Research and Development Agency (Grant No. APVV-22-0291) (M. Balaz), and VEGA (Grant No. 2/0102/23) (M. Balaz). J.M.A. Alcantara was supported by the Spanish Ministry of Education (Ref. FPU15/04059), by MICIU/AEI/ 10.13039/501100011033, and by "ESF+" (Ref. RYC2023-045700-I).

## Author contributions

Conceptualization: C. Wolfrum and C. Wu. Methodology: C. Wu, T.W., and M. Balaz. Investigation: C. Wu, T.W., F.L., A.K.S., T.D., L.D., R.K., S. Kostidis, M.G., L.B., I.S., G.C., S.C., A. Giordano, and M. Balaz. Visualization: C. Wu, T.W., and M. Balaz. Human sample acquisition and processing: J. M.A.A., J.R.R., and M. Blüher. Bioinformatics analyses: A. Ghosh and F.N. (LOBB), V.G., and L.A.N. (ACTIBATE). Comments: A.K.S., B.D., S. Chourasia, S. Kleiner, B.S.H., J.R.R., A.P., M. Balaz, A. Gross, and H.N. Funding acquisition: C. Wolfrum. Supervision: C. Wolfrum (overall), S. Kleiner, B.S.H., A.P., H.N., M. Blüher (LOBB human study), J.R.R. (ACTIBAT human study). C. Wu, M. Balaz, and C. Wolfrum prepared the original draft of the manuscript; A.K.S., S. Kleiner, A.P., and H.N. provided substantial revisions. All authors contributed to editing and reviewing the manuscript.

## Funding

## Competing interests

S.K., B.S.H., A.P., and H.N. are full employees of Boehringer Ingelheim Pharma GmbH & Co. KG, 88397 Biberach an der Riss, Germany. The remaining authors declare no competing interests.

## Additional information

[1]Institute of Food, Nutrition and Health, ETH Zürich, Schwerzenbach, Switzerland. [2]Department of Experimental and Clinical Medicine and Center of Obesity, Marche Polytechnic University, Ancona, Italy. [3]Center for Proteomics and Metabolomics, Leiden University Medical Center, Leiden, The Netherlands. [4]Biomedical Research Center, Slovak Academy of Sciences, Bratislava, Slovakia. [5]Laboratory of Systems Biology and Genetics, Institute of Bioengineering, School of Life Sciences, Ecole Polytechnique Fédérale de Lausanne (EPFL), Lausanne, Switzerland. [6]Swiss Institute of Bioinformatics, Lausanne, Switzerland. [7]Department of Immunology and Regenerative Biology, Weizmann Institute of Science, Rehovot, Israel. [8]Cardiometabolic Diseases Research Department, Boehringer Ingelheim Pharma GmbH & Co. KG, Biberach an der Riss, Germany. [9]PROFITH (PROmoting FITness and Health through Physical Activity) Research Group, Department of Physical Education and Sport, Faculty of Sport Sciences, University of Granada, Granada, Spain. [10]Centro de Investigación Biomédica en Red: Fisiopatología de la Obesidad y Nutrición (CIBEROBN), Madrid, Spain. [11]Department of Education, Faculty of Education Sciences, University of Almería, Almería, Spain. [12]Helmholtz Institute for Metabolic, Obesity and Vascular Research (HI-MAG) of the Helmholtz Zentrum München at the University of Leipzig, Leipzig, Germany. [13]Medical Department III - Endocrinology, Nephrology, Rheumatology, University of Leipzig Medical Center, Leipzig, Germany. [14]Department of Animal Physiology and Ethology, Faculty of Natural Sciences, Comenius University, Bratislava, Slovakia. [15]Nanyang Technological University (NTU), Singapore, Singapore. ✉e-mail: christian.wolfrum@ntu.edu.sg

