## [Transparent Peer Review file · Nature Communications]

MTCH2 modulates CPT1 activity to regulate lipid metabolism of adipocytes

Corresponding Author: Professor Christian Wolfrum

Version 0:

Reviewer comments:

Reviewer #1

(Remarks to the Author)

In this manuscript the authors did a comparison of human and mouse adipose tissue gene expression and with establishing the DEGs up and down, they then performed some functional tests of the identified DEGs of interest and found mitochondrial carrier homolog-2 (Mtch2) to be an interesting protein to further explore. This gene and protein have been studied in the past and associated with a number of disease processes, many of which have links to various cancers. One in particular (PMID: 36264797, 2022) showed this protein to be a mitochondrial outer membrane associated protein and functions as a mitochondrial 'insertase' such that it facilitates the insertion of certain alpha-helical proteins into the mitochondrial membrane, which the authors note. Given the authors' interest in adipose tissue and metabolic homeostasis/obesity, they found in the literature that its increased expression is associated with human obesity. Yet in their studies analyzing several mouse white and brown adipose depots its expression was higher in the beige or brown cells; somewhat confusing.

Some studies of Mtch2 in other cell types observe changes in certain metabolites or control of mitochondrial fusion/fission, and a reported interaction with carnitine palmitoyl transferase-1 (Cpt1; au ref 23). In this work the authors performed 'knock-down' (KD) experiments in human and mouse cells (hMADS and immortalized brown adipocytes (iBAs), respectively). They observed increased respiration and mitochondrial oxidation of fuels in the KD cells and surprisingly, even when Ucp1 expression was also deleted in these cell models, this phenotype persisted. Examining what is the fuel preference of these cells they found an increased demand for fatty acids.

Since a recent publication made note of 'inosine' being released from dying/apoptotic cells (au ref 40; original publication PMID: 35790189; 2022), this was investigated as a potential mechanism behind the effects of Mtch2 ablation – however there was no evidence of apoptosis in the KD cells vs the WT, seeming to rule out that process. They also performed extensive biochemical analyses to test various possible metabolic perturbations, many that have been reported, but they did not detect differences in any of metabolites they screened for, including creatines. They also did not detect differences in mitochondrial dynamics (gene expression and by electron microscopy of mitochondria), or alterations in mitochondrial electron transport complex content. A mouse model of inducible 'knock-out' of Mtch2 was generated (AdipoQ-CreERT2 Mtch2). These KO mice on chow diet showed not differences, except a modest increase in energy expenditure that could not be explained by body weight or composition. On high-fat diet the KO mice did show a reduced body weight, which was fat weight, and improvements in glucose tolerance; perhaps a consequence of the lower amount of body fat.

Mechanistically they did find a direct interaction of Mtch2 with Cpt1a and Cpt1b, as found by others (au ref 23). The experiments performed by the authors here to confirm this interaction were rigors and convincing. Since a known regulator of Cpt1 activity is malonyl-CoA, they investigated whether the loss of Mtch2 affected the amount of malonyl-CoA, but there was no difference. It would be interesting to know what regions of the two proteins interact and are those motifs found in other mito proteins. Information does exist as to some functional domains of Cpt1 and it is possible to genetically/biochemically manipulate Cpt1 structure as well as Mtch2 structure, but this was not attempted. The authors state that such work is hampered by a lack of a crystal structure of Cpts. This reviewer is less convinced that this derails such experiments. So at this point the authors can show an effect of deletion/KD of Mtch2 in terms of mitochondrial fatty acid respiration, but the mechanism behind this – other than an interaction with Cpt1 – is absent.

The authors refer to ref 50 – a BioRxiv preprint indicated to be accepted, and now published: EMBO J. 2025 Jan 3. doi: 10.1038/s44318-024-00335-7. PMID: 39753955. This very comprehensive work shows Mtch2 deletion results in very high ATP demand. Extensive metabolomics and lipidomics were done in HeLa cells that are WT, Mtch2 KO or KO-rescued. They show a clear decrease in the ADP/ATP ratio. Also this EMBO J work shows the KO cells have a decrease in

membrane lipids that in cells is used either for storage (e.g. TAG) or oxidation and energy production. The lipidomics done in this EBMO J paper were rigorous and detailed (performed by arguably the best person in the field for such work; Han), and their data clearly show an increase in TAG storage in cells. I think the authors of the current NCOMM manuscript should investigate this possibility in their own studies. It is possible that for some reason the increased need of fuel oxidation is that membrane lipids are being stored and there is a need to replenish these.

As for the significance of this work with an eye toward human biology, it raises a few important questions. If Mtch2 only works in brown adipocytes, and in humans there are relatively few, and the point has been discussed that it is questionable for brown adipocytes to make a difference in human EE (PMID: 38871497 - - this review focuses on FDG (glucose) uptake into human UCP1+ BAT...so with that caveat), what is the real relevance here, other than some understanding of a biochemical process in adipocytes that can be ascribed to Mtch2? Since the author's studies with cell models find increased mitochondrial respiration without UCP1 (experiments not done in cells from the mice in which KD of Ucp1 could be performed), the authors refer to adipose tissue UCP1-independent mechanisms including creatine cycling. However, such statements should be balanced by the view of outstanding investigators of mitochondrial energetics (PMID: 36624158), that bioenergetically the contribution of creatine cycling to EE is strongly questioned. The authors of this NCOMM ms do note the lack of clarity as to the importance of this pathway in humans (their ref 62), but they should include the reference provided here (PMID: 36624158).

Given that Mtch2 is not a 'new' protein involved in mitochondrial metabolism, and that the interaction with Cpt1 is not new, the value of the work is that it has been done in a relevant metabolic tissue – adipose – and comparing mouse and human and finding similarities in the expression of Mtch2. However, in the absence of a more rigorous mechanistic dissection of the functional importance of this interaction the work does not break significant new ground in the topic.

Reviewer #2

(Remarks to the Author)

The authors perform an extensive transcriptome analysis of numerous adipose models, and identify over 600 genes that are in common between the paired brown adipose and white adipose models. Of these candidates, they perform knockdown on 110 further curated candidates. Of these, MTCH2 was the candidate with the greatest averaged effects across cAMP-stimulated and maximal respiration upon knockdown. They then demonstrate that MTCH2 expression is correlated with obesity and several metabolism parameters in humans. They then demonstrate in human culture models that MTCH2 affects mitochondrial respiration in a UCP1-independent fashion. They study an inducible adipocyte knockout model and demonstrate these mice have increased energy expenditure, weigh less due to decreased fat mass and have improved glucose tolerance. These effects are not accompanied by overt mitochondrial morphologic differences, but with increased mitochondrial capacity for fatty acid oxidation, due to increased CPT1 activity. Last, co-IP and PLA assays are used to demonstrate interaction between MTCH2 and CPT1. A model is then proposed that MTCH2 downregulates beta oxidation by blocking CPT1 transport. This manuscript is based off of a well designed screen with extensive mechanistic follow-up studies. The interaction between MTCH2 and CPT1 could be more convincingly supported, but the data presented are consistent with the conclusions. One notable omission in the manuscript is RER from the metabolic cages (unless I just missed it). Given the model presented, an effect on RER would be expected.

Reviewer #3

(Remarks to the Author)

This manuscript investigates the role of MTCH2 in adipose tissue. Adipocyte-specific ablation of MTCH2 improves mitochondrial function and whole-body energy expenditure. Importantly, MTCH2 expression correlates with obesity in humans. This is a very interesting study, but a few points should be addressed before publication.

Major points:

- Analysis of MTCH2 expression should include Western blotting (fig. 1f).
- Expression of MTCH2 is rather high in iBAT. Is MTCH2 expression altered in the different murine adipose tissue depots during cold, thermoneutrality or obesity?
- Knockdown of MTCH2 results in reduced UCP1 expression. Has the loss of MTCH2 any significant effect on general differentiation of human adipocytes (hMADS)? Data on the expression of adipogenic and thermogenic marker genes would be helpful.
- Fig 7, the take home message is not clear, the different colors of the arrows are not defined, (e.g. the red arrows from acyl-coa and carnitine in the right part versus black in the left part?).

Minor points:

- line 19: "these three datasets revealed 679 overlapping DEGs (FDR<0.05; Fig. 1a and Supplementary Table 4)."
Overlapping between which data sets?

Version 1:

Reviewer comments:

Reviewer #1

(Remarks to the Author)

My concerns have been adequately addressed.

Reviewer #2

(Remarks to the Author)

The authors have addressed the reviewer's concerns. Excellent work!

REVIEWER COMMENTS

Reviewer #1 (Remarks to the Author):

In this manuscript the authors did a comparison of human and mouse adipose tissue gene expression and with establishing the DEGs up and down, they then performed some functional tests of the identified DEGs of interest and found mitochondrial carrier homolog-2 (Mtch2) to be an interesting protein to further explore. This gene and protein have been studied in the past and associated with a number of disease processes, many of which have links to various cancers. One in particular (PMID: 36264797, 2022) showed this protein to be a mitochondrial outer membrane associated protein and functions as a mitochondrial ‘insertase’ such that it facilitates the insertion of certain alpha-helical proteins into the mitochondrial membrane, which the authors note. Given the authors’ interest in adipose tissue and metabolic homeostasis/obesity, they found in the literature that its increased expression is associated with human obesity. Yet in their studies analyzing several mouse white and brown adipose depots its expression was higher in the beige or brown cells; somewhat confusing.

Response: Thank you for your comments and apologies for the confusion. MTCH2 is expressed in both white and brown adipocytes, with notably higher expression in beige and brown adipocytes. This is largely due to MTCH2 being a mitochondrial outer membrane protein, and its expression is associated with the level of mitochondrial activity and catabolism. Regarding its association with human obesity, increased MTCH2 expression has been observed in both subcutaneous and visceral white adipose tissue, suggesting its relevance to white adipose tissues in the context of obesity. We have clarified this in the revised manuscript.

Some studies of Mtch2 in other cell types observe changes in certain metabolites or control of mitochondrial fusion/fission, and a reported interaction with carnitine palmitoyl transferase-1 (Cpt1; au ref 23). In this work the authors performed ‘knock-down’ (KD) experiments in human and mouse cells (hMADS and immortalized brown adipocytes (iBAs), respectively). They observed increased respiration and mitochondrial oxidation of fuels in the KD cells and surprisingly, even when Ucp1 expression was also deleted in these cell models, this phenotype persisted. Examining what is the fuel preference of these cells they found an increased demand for fatty acids. Since a recent publication made note of ‘inosine’ being released from dying/apoptotic cells (au ref 40; original publication PMID: 35790189; 2022), this was investigated as a potential mechanism behind the effects of Mtch2 ablation – however there was no evidence of apoptosis in the KD cells vs the WT, seeming to rule out that process. They also performed extensive biochemical analyses to test various possible metabolic perturbations, many that have been reported, but they did not detect differences in any of metabolites they screened for, including creatines. They also did not detect differences in mitochondrial dynamics (gene expression and by electron microscopy of mitochondria), or alterations in mitochondrial electron transport complex content. A mouse model of inducible ‘knock-out’ of Mtch2 was generated (AdipoQ-CreERT2 Mtch2). These KO mice on chow diet showed not differences, except a modest increase in energy expenditure that could not be explained by body weight or composition. On high-fat diet the KO mice did show a reduced body weight, which was fat weight, and improvements in glucose tolerance; perhaps a consequence of the lower amount of body fat.

Mechanistically they did find a direct interaction of Mtch2 with Cpt1a and Cpt1b, as found by others (au ref 23). The experiments performed by the authors here to confirm this interaction were rigors and convincing. Since a known regulator of Cpt1 activity is malonyl-CoA, they investigated whether the loss of Mtch2 affected the amount of malonyl-CoA, but there was no difference. It would be interesting to know what regions of the two proteins interact and are those motifs found in other mito proteins. Information does exist as to some functional domains of Cpt1 and it is possible to

genetically/biochemically manipulate Cpt1 structure as well as Mtch2 structure, but this was not attempted. The authors state that such work is hampered by a lack of a crystal structure of Cpts. This reviewer is less convinced that this derails such experiments. So at this point the authors can show an effect of deletion/KD of Mtch2 in terms of mitochondrial fatty acid respiration, but the mechanism behind this – other than an interaction with Cpt1 – is absent.

Response: Thank you for the comments and suggestions. To investigate which region of MTCH2 is involved in this interaction, we generated a siRNA-resistant FLAG-MTCH2 wild-type (WT) construct using a lentiviral system. We then produced seven truncation mutants (M1–M7) on the WT backbone based on published MTCH2 structure (Extended Data Fig. 7c). All mutations were overexpressed in HEK293T cells to assess MTCH2-CPT1 interaction through immunoprecipitation assays. All mutations were subsequently introduced into iBAs via lentiviral infection, followed by siRNA-mediated knockdown of endogenous MTCH2 to evaluate their effects on CPT1 activity. All mutants were successfully expressed and localized to mitochondria in HEK293T cells, except for the N-terminal truncation mutant (M1) (Extended Data Fig. 7d), suggesting that the N-terminus is essential for proper protein synthesis or stability. Notably, MTCH2-CPT1 interaction was preserved in all other mutants (M2–M6) except for M7 (Extended Data Fig. 7e). Moreover, only M7 construct was able to restore the CPT1 activity induced by MTCH2 depletion (Extended Data Fig. 7f). These findings suggest that the C-terminus of MTCH2 is essential for both its interaction with CPT1 and the regulation of CPT1 activity. We have revised our text on Page 12 (Line 326-336) and related results were shown in Extended Data Fig.7c-f.

In addition, we performed bioinformatic analyses to identify other mitochondrial proteins with sequence similarity to the C-terminus of MTCH2, aiming to uncover additional candidates that might interact with CPT1. Among the hits, MTCH1 showed the highest similarity, with 50% sequence identity. However, our previous experiments testing the effects of MTCH1 on cellular respiration in human adipocytes showed no significant impact (Extended Data Fig. 7g), suggesting that MTCH2 interact with CPT1 through distinct and functionally specific regions not shared with MTCH1, or potentially due to differences in their three-dimensional protein structures. We have also clarified this in the revised manuscript on Page 12-13 (line 336-343).

The authors refer to ref 50 – a BioRxiv preprint indicated to be accepted, and now published: EMBO J. 2025 Jan 3. doi: 10.1038/s44318-024-00335-7. PMID: 39753955. This very comprehensive work shows Mtch2 deletion results in very high ATP demand. Extensive metabolomics and lipidomics were done in HeLa cells that are WT, Mtch2 KO or KO-rescued. They show a clear decrease in the ADP/ATP ratio. Also this EMBO J work shows the KO cells have a decrease in membrane lipids that in cells is used either for storage (e.g. TAG) or oxidation and energy production. The lipidomics done in this EBMO J paper were rigorous and detailed (performed by arguably the best person in the field for such work; Han), and their data clearly show an increase in TAG storage in cells. I think the authors of the current NCOMM manuscript should investigate this possibility in their own studies. It is possible that for some reason the increased need of fuel oxidation is that membrane lipids are being stored and there is a need to replenish these.

Response: Thank you for your comments. To explore this possibility, we investigated the levels of membrane and storage lipids in human adipocytes (hMADS) upon MTCH2 depletion. Absolute concentration analysis revealed that major lipid levels remained unchanged; however, phosphatidylserine (PS) levels were significantly increased (Extended Data Fig. 6f,g). In contrast, lipid composition percentage analysis revealed a significantly decrease in phosphatidic acid (PA) and storage lipid cholesteryl esters (CE) (Extended Data Fig. 6h,i). The reduction in PA may be linked to MTCH2's role as a mitochondrial insertase or scramblase, or alternatively, PA may be used for oxidation to meet increased energy demands. The decrease in CE might result from impaired cholesterol metabolism,

reduced cholesterol levels are also reported in HeLa cells. Overall, MTCH2 depletion induced mild alterations in membrane and storage lipid levels in mature hMADS adipocytes, which was less pronounced compared to those observed in HeLa cells. This discrepancy may be attributed to cell type-specific differences in lipid metabolism. We have integrated these deliberations into the revised manuscript on Page 11 (line 295-300).

As for the significance of this work with an eye toward human biology, it raises a few important questions. If *Mtch2* only works in brown adipocytes, and in humans there are relatively few, and the point has been discussed that it is questionable for brown adipocytes to make a difference in human EE (PMID: 38871497 - - this review focuses on FDG (glucose) uptake into human UCP1+ BAT...so with that caveat), what is the real relevance here, other than some understanding of a biochemical process in adipocytes that can be ascribed to *Mtch2*? Since the author's studies with cell models find increased mitochondrial respiration without UCP1 (experiments not done in cells from the mice in which KD of *Ucp1* could be performed), the authors refer to adipose tissue UCP1-independent mechanisms including creatine cycling. However, such statements should be balanced by the view of outstanding investigators of mitochondrial energetics (PMID: 36624158), that bioenergetically the contribution of creatine cycling to EE is strongly questioned. The authors of this NCOMM ms do note the lack of clarity as to the importance of this pathway in humans (their ref 62), but they should include the reference provided here (PMID: 36624158).

Response: Thank you for raising these important points. Our data indicated MTCH2 is expressed in both white and brown adipocytes, with notably higher expression in beige and brown adipocytes. This is due to MTCH2 being a mitochondrial outer membrane protein, and its expression correlates with mitochondrial activity and catabolism. Regarding its association with human obesity, increased *MTCH2* expression has been observed in both subcutaneous and visceral white adipose tissue (WAT), suggesting its relevance to WAT in the context of obesity. Moreover, we observed *MTCH2* expression in human scWAT was negatively correlated with energy expenditure. Therefore, MTCH2 functions not limited in human BAT but also in WAT, which is the most abundant adipose tissue in individuals with obesity, reinforcing the translational relevance of our findings.

To study the relation with UCP1, we isolated the stromal vascular fraction from control and *Mtch2^{fl/fl}Adip-CreERT2* mouse iBAT, followed by culture and differentiation into mature adipocytes. Subsequently, the adipocyte were treated with 4-hydroxy-tamoxifen to induce *Mtch2* knockout and *Ucp1* was knocked down using siRNA. The results showed that the effects of *Mtch2* on cellular respiration also persisted in *Ucp1* ablated primary adipocytes (Extended Data Fig. 3e), consistent with our observation in immortalized brown adipocytes (Extended Data Fig. 3d). These findings further support that MTCH2 regulates mitochondrial respiration independent of UCP1. We have also revised the text to include these data on Page 8 (Line 204-205).

We are aware of the ongoing debate regarding the contribution of creatine cycling to energy expenditure, and have added the reference reviewer provided (PMID: 36624158) to further clarify this point, as shown on Page 15 (Line 405 for reference 55).

Given that *Mtch2* is not a 'new' protein involved in mitochondrial metabolism, and that the interaction with *Cpt1* is not new, the value of the work is that it has been done in a relevant metabolic tissue – adipose – and comparing mouse and human and finding similarities in the expression of *Mtch2*. However, in the absence of a more rigorous mechanistic dissection of the functional importance of this interaction the work does not break significant new ground in the topic.

Response: Thank you for your thoughtful comments. We fully agree that MTCH2 is not a novel factor in mitochondrial metabolism, and that its interaction with CPT1 has been reported. However, our study provides new insights by demonstrating that the MTCH2-CPT1 interaction directly regulates CPT1

activity, thereby contributing to fatty acid oxidation and overall energy homeostasis—a functional consequence that, to our knowledge, has not been previously described.

Importantly, in this revision we identified the C-terminal region of MTCH2 as essential for both its interaction with CPT1 and the regulation of its activity, offering new mechanistic understanding. Moreover, our findings extend MTCH2 function beyond brown adipose tissue (BAT) to white adipose tissue (WAT), which is the predominant adipose depot in individuals with obesity. With the ongoing debate surrounding the presence and functional significance of BAT in adult humans, our observation that MTCH2 expression in human subcutaneous WAT negatively correlates with energy expenditure further reinforces the translational relevance of our study.

We believe these mechanistic and physiological insights will help advance our understanding of how MTCH2 regulates adipose tissue metabolism, thereby contributing to energy homeostasis and holding potential relevance to obesity and related metabolic disorders.

Reviewer #2 (Remarks to the Author):

The authors perform an extensive transcriptome analysis of numerous adipose models, and identify over 600 genes that are in common between the paired brown adipose and white adipose models. Of these candidates, they perform knockdown on 110 further curated candidates. Of these, MTCH2 was the candidate with the greatest averaged effects across cAMP-stimulated and maximal respiration upon knockdown. They then demonstrate that MTCH2 expression is correlated with obesity and several metabolism parameters in humans. They then demonstrate in human culture models that MTCH2 affects mitochondrial respiration in a UCP1-independent fashion. They study an inducible adipocyte knockout model and demonstrate these mice have increased energy expenditure, weigh less due to decreased fat mass and have improved glucose tolerance. These effects are not accompanied by overt mitochondrial morphologic differences, but with increased mitochondrial capacity for fatty acid oxidation, due to increased CPT1 activity. Last, co-IP and PLA assays are used to demonstrate interaction between MTCH2 and CPT1. A model is then proposed that MTCH2 downregulates beta oxidation by blocking CPT1 transport. This manuscript is based off of a well designed screen with extensive mechanistic follow-up studies. The interaction between MTCH2 and CPT1 could be more convincingly supported, but the data presented are consistent with the conclusions. One notable omission in the manuscript is RER from the metabolic cages (unless I just missed it). Given the model presented, an effect on RER would be expected.

Response: Thank you for comments and for raising this important point. The RER data are shown in Extended Data Fig. 3g. And indeed we observed a reduction in RER in *Mtch2* KO mice, suggesting a shift toward increased fat utilization as energy source, which is consistent with our data that loss of MTCH2 shifts substrate utilization towards fatty acids and upregulates fatty acid oxidation. We have also revised our text to include this information on Page 8 (Line 213-214).

Reviewer #3 (Remarks to the Author):

This manuscript investigates the role of MTCH2 in adipose tissue. Adipocyte-specific ablation of MTCH2 improves mitochondrial function and whole-body energy expenditure. Importantly, MTCH2 expression correlates with obesity in humans. This is a very interesting study, but a few points should be addressed before publication.

Major points:

-Analysis of MTCH2 expression should include Western blotting (fig. 1f).

Response: Thank you for the comments and suggestions. We analyzed MTCH2 protein levels in paired supraclavicular BAT and adjacent subcutaneous WAT (scWAT) biopsies of seven volunteers. And the results confirmed that MTCH2 expression is higher in iBAT than that in scWAT (Extended Data Fig. 1a), which is consistent with gene expression data (Fig. 1f). We have also revised the text to include this new data on Page 6 (Line 150-151).

-Expression of MTCH2 is rather high in iBAT. Is MTCH2 expression altered in the different murine adipose tissue depots during cold, thermoneutrality or obesity?

Response: Thank you for your comments and for raising this important point. We performed qPCR to examine *Mtch2* expression in adipose tissues under chow, high fat diet (HFD, 6 weeks) and thermoneutrality conditions. The results showed that *Mtch2* expression increased following HFD and decreased under thermoneutrality conditions across all three murine adipose tissue depots, including inguinal and epididymal white adipose tissues (iWAT and eWAT), and interscapular brown adipose tissue (iBAT) (Extended Data Fig. 1f). We also checked published datasets to further assess the effects of cold exposure. The previous study results show that cold exposure induces *Mtch2* expression across these three adipose depots, although the increase in iBAT does not reach statistical significance ($p=0.0517$) (PMID: 37142619). Consistently, *Mtch2* expression decreases under thermoneutrality condition (PMID: 29657031) (Extended Data Fig. 1g). Collectively, these data indicate that *Mtch2* expression is upregulated by HFD and cold exposure, and downregulated under thermoneutrality condition, which might be largely due to MTCH2 being a mitochondrial outer membrane protein, and its expression is associated with the level of mitochondrial activity and catabolism. We have also revised the text to include this new data on Page 6 (Line 160-163).

-Knockdown of MTCH2 results in reduced UCP1 expression. Has the loss of MTCH2 any significant effect on general differentiation of human adipocytes (hMADS)? Data on the expression of adipogenic and thermogenic marker genes would be helpful.

Response: Thank you for raising this important point. We examined the effects of *MTCH2* knockdown on the expression of both adipogenic and thermogenic marker genes, including *FABP4*, *ADIPOQ*, *PPARG*, *PPARGC1A* and *CIDEA*. The results showed that *MTCH2* knockdown did not significantly alter the expression of these marker genes (Extended Data Fig. 3b). Therefore, we conclude that *MTCH2* knockdown does not affect the general differentiation of hMADS, consistent with the minimal changes observed in these marker gene expressions. We have integrated these deliberations into the revised manuscript on Page 7 (line 196-198).

-Fig 7, the take home message is not clear, the different colors of the arrows are not defined, (e.g. the red arrows from acyl-coa and carnitine in the right part versus black in the left part?).

Response: Thank you for your comments. We appreciate your observation regarding the clarity of Fig. 7. We have revised the figure legend on Page 18 to include definitions for the different arrow colors and to clarify the overall take-home message of the figure. We hope the updated version provides a clearer representation of the proposed model.

Minor points:

-line119: “these three datasets revealed 679 overlapping DEGs (FDR<0.05; Fig. 1a and Supplementary Table 4).” Overlapping between which data sets?

Response: Thank you for your comments and apologize for any confusion. We first identified differentially expressed genes (DEGs) from each of the following datasets, (1) human BAT vs. scWAT biopsies (2680DEGs), (2) white vs. beige hMADS adipocytes (5004 DEGs), and (3) murine iBAT vs. iWAT populations (7265 DEGs). Then we overlapped these three DEGs datasets to identify potential genes involved in regulating mitochondrial function. We have revised our text on Page 5 (Line 118-120) to clearly define the three datasets used for overlap. We hope this helps clarify this information.